# Working memory capacity of crows and monkeys arises from similar neuronal computations

Lukas Alexander Hahn[1]*, Dmitry Balakhonov[1]*, Erica Fongaro[1], Andreas Nieder[2], Jonas Rose[1]

[1]Neural Basis of Learning, Institute of Cognitive Neuroscience, Faculty of Psychology, Ruhr University Bochum, Bochum, Germany; [2]Animal Physiology, Institute of Neurobiology, University of Tübingen, Tübingen, Germany

**Abstract** Complex cognition relies on flexible working memory, which is severely limited in its capacity. The neuronal computations underlying these capacity limits have been extensively studied in humans and in monkeys, resulting in competing theoretical models. We probed the working memory capacity of crows (*Corvus corone*) in a change detection task, developed for monkeys (*Macaca mulatta*), while we performed extracellular recordings of the prefrontal-like area nidopallium caudolaterale. We found that neuronal encoding and maintenance of information were affected by item load, in a way that is virtually identical to results obtained from monkey prefrontal cortex. Contemporary neurophysiological models of working memory employ divisive normalization as an important mechanism that may result in the capacity limitation. As these models are usually conceptualized and tested in an exclusively mammalian context, it remains unclear if they fully capture a general concept of working memory or if they are restricted to the mammalian neocortex. Here, we report that carrion crows and macaque monkeys share divisive normalization as a neuronal computation that is in line with mammalian models. This indicates that computational models of working memory developed in the mammalian cortex can also apply to non-cortical associative brain regions of birds.

*For correspondence:
lukas.hahn@ruhr-uni-bochum.de
(LAlexanderH);
balakhonov.ds@gmail.com (DB)

Competing interest: The authors declare that no competing interests exist.

## Editor's evaluation

In this study, Hahn et al., taught crows to perform a multi-item working memory task designed to mimic traditional monkey tasks. Using a combination of behavior and electrophysiology, the authors convincingly show that the neural mechanisms that limit working memory capacity in primates also limit working memory capacity in crows. Such cross-species comparisons are fundamental to understanding the computational constraints that are placed on cognition and the brain.

## Introduction

Working memory (WM) can hold information for a short period of time to allow further processing in the absence of sensory input (*Cowan, 2017*; *Oberauer et al., 2018*). By bridging this gap between the immediate sensory environment and behavior, WM is a cornerstone for complex cognition. It is a very flexible memory system, yet severely limited in its capacity. While this capacity is often seen as a general cognitive bottleneck, for simple stimuli, like colors, the capacity is very similar between humans, monkeys, and crows (*Balakhonov and Rose, 2017*; *Buschman et al., 2011*; *Cowan, 2001*; *Luck and Vogel, 1997*). Different models have been proposed to conceptualize how this capacity limit arises. This work motivated many psychophysical and electrophysiological experiments that in

**eLife digest** Working memory is the brain's ability to temporarily hold and manipulate information. It is essential for carrying out complex cognitive tasks, such as reasoning, planning, following instructions or solving problems. Unlike long-term memory, information is not stored and recalled, but held in an accessible state for brief periods. However, the capacity of working memory is very limited. Humans, for example, can only hold around four items of information simultaneously.

There are various competing theories about how this limitation arises from the network of neurons in the brain. These models are based on studies of humans and other primates. But memory limitations are not exclusive to mammals. Indeed, the working memory of some birds, such as crows, has a similar capacity to humans despite the architecture of their brains being very different to mammals. So, how do brains with such distinct structural differences produce working memories with similar capacities?

To investigate, Hahn et al. probed the working memory of carrion crows in a change detection task developed for macaque monkeys. Crows were trained to memorize varying numbers of colored squares and indicate which square had changed after a one second delay when the screen went blank. While the crows performed the task, Hahn et al. measured the activity of neurons in an area of the brain equivalent to the prefrontal cortex, the central hub of cognition in mammals.

The experiments showed that neurons in the crow brain responded to the changing colors virtually the same way as neurons in monkeys. Hahn et al. also noticed that increasing the number of items the crows had to remember affected individual neurons in a similar fashion as had previously been observed in monkeys.

This suggests that birds and monkeys share the same central mechanisms of, and limits to, working memory despite differences in brain architecture. The similarities across distantly related species also validates core ideas about the limits of working memory developed from studies of mammals.

turn led to a spectrum of more refined models of WM (*Ma et al., 2014*). 'Discrete models' of WM argue that a fixed number of items can be stored. Once this capacity is reached, an additional item can only be maintained if it replaces a previous item (*Awh et al., 2007*; *Fukuda et al., 2010*; *Luck and Vogel, 1997*; *Vogel and Machizawa, 2004*; *Zhang and Luck, 2008*). 'Continuous models' describe WM as a flexible resource that is allocated to individual items. A minimum amount of this resource has to be allocated to each item for successful retention, thereby resulting in a capacity limit (*Bays and Husain, 2008*; *van den Berg et al., 2012*; *Wilken and Ma, 2004*). On the neurophysiological level, models of WM capacity suggest that interference between memory representations ('*items*') within the neuronal network is a source of information loss and capacity limitation (*Bouchacourt and Buschman, 2019*; *Lundqvist et al., 2016*; *Lundqvist et al., 2011*; *Schneegans et al., 2020*). Interference may arise due to divisive normalization that appears as competition between items, related to oscillatory dynamics (*Lundqvist et al., 2016*; *Lundqvist et al., 2011*), WM flexibility (*Bouchacourt and Buschman, 2019*), and neuronal information sampling (*Schneegans et al., 2020*). Divisive normalization is a computational principle that acts upon neurons when presenting multiple stimuli simultaneously, it normalizes neuronal responses by creating 'a ratio between the response of an individual neuron and the summed activity of a pool of neurons' (*Carandini and Heeger, 2011*, p. 51). An effect related to divisive normalization can be observed when two stimuli are presented either individually or simultaneously within the receptive field of a visual sensory neuron. The neuron's firing rate when the stimuli are presented simultaneously becomes normalized by the populations' responses to each individual stimulus (*Carandini et al., 1997*; *Heeger, 1992*). This effect also occurs in relation to attentive processes and has been formalized into the 'normalization model of attention' (*Reynolds et al., 1999*; *Reynolds and Heeger, 2009*). Normalization of neuronal responses is commonly observed in many species throughout the animal kingdom, not just in sensory, but also in cognitive domains (*Carandini and Heeger, 2011*). Investigations into WM capacity and model predictions focus mostly on humans and monkeys. By extending this work to include birds, one can gain a unique comparative perspective. Crows have a similar limit in WM capacity and neuronal correlates of WM are comparable to monkeys' (*Balakhonov and Rose, 2017*; *Nieder, 2017*). But while the neuronal architecture of sensory areas is similar between birds and mammals, higher associative areas, critical for WM, do not

share a common architecture between the species (*Stacho et al., 2020*). Therefore, an outstanding question is whether modern models of WM such as the 'flexible model' capture WM capacity in general, or if their predictions (e.g., divisive normalization) are confined to the mammalian neocortex. To resolve this, it is crucial to investigate the avian brain to understand how its different organization can produce such similar behavioral and neurophysiological results. While the neuronal correlates of WM maintenance in birds have been investigated in some detail (*Diekamp et al., 2002a*; *Hartmann et al., 2018*; *Rinnert et al., 2019*; *Rose and Colombo, 2005*; *Veit et al., 2014*), a neurophysiological investigation of WM capacity limitation is still lacking. The avian forebrain structure, *nidopallium caudolaterale* (NCL) is a critical component of avian WM. The NCL is considered functionally equivalent to the mammalian prefrontal cortex (PFC) (*Güntürkün and Bugnyar, 2016*; *Nieder, 2017*), as it receives projections from all sensory modalities (*Kröner and Güntürkün, 1999*), projects to premotor areas (*Kröner and Güntürkün, 1999*), and is a target of dopaminergic innervation (*Waldmann and Güntürkün, 1993*). To investigate the neurophysiology of WM capacity in birds, we adopted a task design developed for monkeys (*Buschman et al., 2011*) to use it with carrion crows (*Corvus corone*). Our animals were trained to memorize an array of colors and to indicate which color had changed after a short memory delay, while we performed extracellular recordings of individual neurons in the NCL using multi-channel probes. We expected to find a clear correlate of WM representations in NCL neurons and a load-dependent response modulation based on divisive normalization of neuronal responses. This would allow us to evaluate if the behavioral WM capacity observations of crows fit a 'discrete' or 'continuous' WM resource model. If the neuronal responses also fit the contemporary neurophysiological models of WM capacity limitations (*Bouchacourt and Buschman, 2019*; *Lundqvist et al., 2016*; *Lundqvist et al., 2011*; *Schneegans et al., 2020*), it would further suggest that crows and monkeys have convergently evolved a similar neurophysiological basis for WM capacity despite a different architecture of the critical forebrain structures.

## Results

### The WM capacity of crows is similar to that of monkeys

The behavioral performance was influenced by the number of colored squares on the screen. It significantly decreased with an increasing number of ipsilateral squares (median performances, load 1: 95.88%, load 2: 78.31%, load 3: 58.21%; Friedman test: $X^2$ = 92.00, p < 0.001, *Figure 1B*). We ran a generalized linear model with ipsilateral load (i.e., load of hemifield where a color changed), contralateral load (i.e., load of hemifield without a color change) and their interaction as predictors for performance ($R^2_{adj}$ 0.78, F(460,456) = 555.00, p < 0.001). We found that the number of ipsilateral colors significantly reduced performance ($\beta_{ipsi}$ –0.177, t(459) = –18.00, p < 0.001), whereas the number of contralateral colors did not ($\beta_{contra}$ –0.021, t(459) = –1.77, p = 0.0772; *Figure 1C*). There was also a significant interaction between ipsilateral and contralateral load ($\beta$ = –0.024, t(458) = –4.28, p < 0.001). We compared this model to a reduced model, where we omitted the non-significant $\beta_{contra}$ and found that this reduced model ($R^2_{adj}$ 0.78, F(460,457) = 828.00, p < 0.001) explained the performance equally as well (|$\Delta$LLR| = 0.0102). Therefore, we conclude that contralateral load by itself did not significantly affect performance. We calculated the capacity $K$ (see Materials and methods) for all full WM loads (i.e., two to five items). The capacity $K$ peaks at four items (mean ± SEM: 3.05±.038, *Figure 1—figure supplement 1*). These observations are very similar to observations made in primates (*Buschman et al., 2011*) and fully reproduce our earlier behavioral findings (*Balakhonov and Rose, 2017*).

### Neurons of the NCL encode the color identity and maintain it in WM

We recorded 362 neurons from the NCL of two crows performing the WM task (delayed change localization). All reported effects were also present in each individual bird (*Figure 3—figure supplements 1–2*), we, therefore, pooled the data for population analysis. A large subset of neurons responded to the presence of a color (i.e., at load 1) by substantially increasing or decreasing their firing rate relative to baseline. This change in firing rate occurred selectively, depending on the presented color either in the sample (*Figure 2A*) or the delay period (*Figure 2—figure supplement 1*). For most neurons, this difference in firing rate between the two possible colors became attenuated when the load increased from one to two colors, and it was further attenuated from two to three colors. To quantify this effect, we calculated the amount of information about the color identity at a neuron's favorite location as

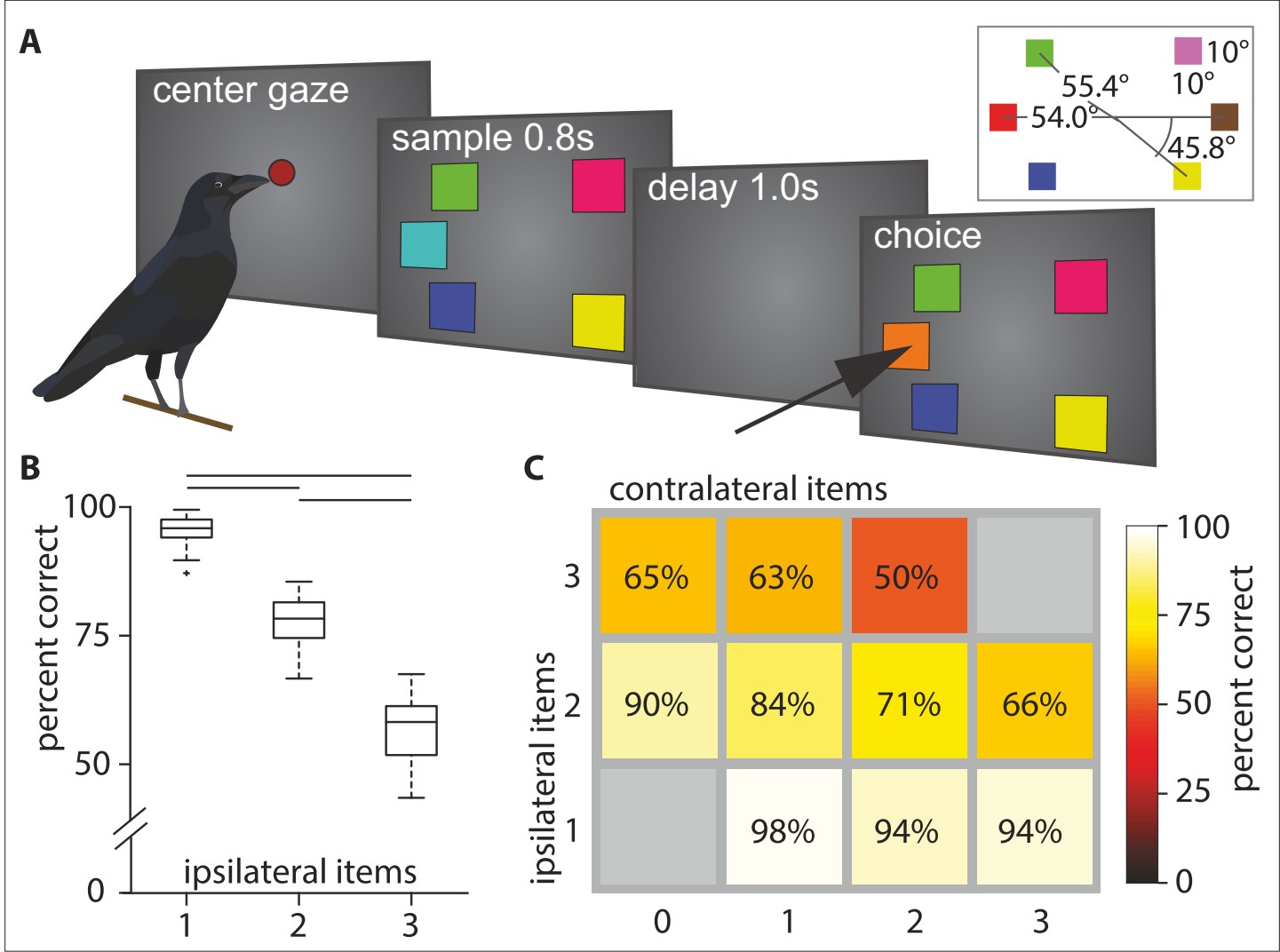

**Figure 1.** Behavioral overview. (**A**) Behavioral paradigm (reproduced from *Balakhonov and Rose, 2017*). The birds had to center and hold their gaze for the duration of the sample and delay period, and subsequently indicate which colored square had changed. (**B**) Boxplot of performance for different ipsilateral loads (i.e., on the side where the change occurred). Horizontal lines indicate significant differences between loads, box indicates the median, 1st, and 3rd quartile (whiskers extend to 1.5 times the inter-quartile range). (**C**) Mean performance matrix for ipsi- and contralateral load combinations (values are rounded to the nearest integer). Additional contralateral items at an ipsilateral load of 1 barely affected performance (bottom row). At higher ipsilateral loads additional contralateral items reduced performance more clearly (middle and top row). Statistical modeling revealed an interaction at these higher loads (see text).

The online version of this article includes the following figure supplement(s) for figure 1:

**Figure supplement 1.** Capacity of crow working memory (WM).

the percent explained variance (PEV) during a memory load of one, two, or three items in bins of 200 ms (see Materials and methods for details). Most neurons did not sustain information about color (measured as a significant PEV, henceforth '*information*') throughout the entire sample or memory delay but rather had shorter periods in which the information was significant (*Figure 2A* bottom).

To better capture the time points when the individual neurons carried information, we performed a hierarchical clustering analysis of the PEV values of the individual neurons at load 1 (see Materials and methods for details). We found a total of seven clusters that were organized into two overarching groups (*Figure 3A*). Group 1 contained neurons (n = 227) that showed peak information during the sample and early delay phase, while group 2 contained neurons (n = 135) that showed peak information during the delay phase. For each neuron, we then calculated if it carried a significant amount of color information by applying a permutation test (for all bins at load 1, see Materials and methods).

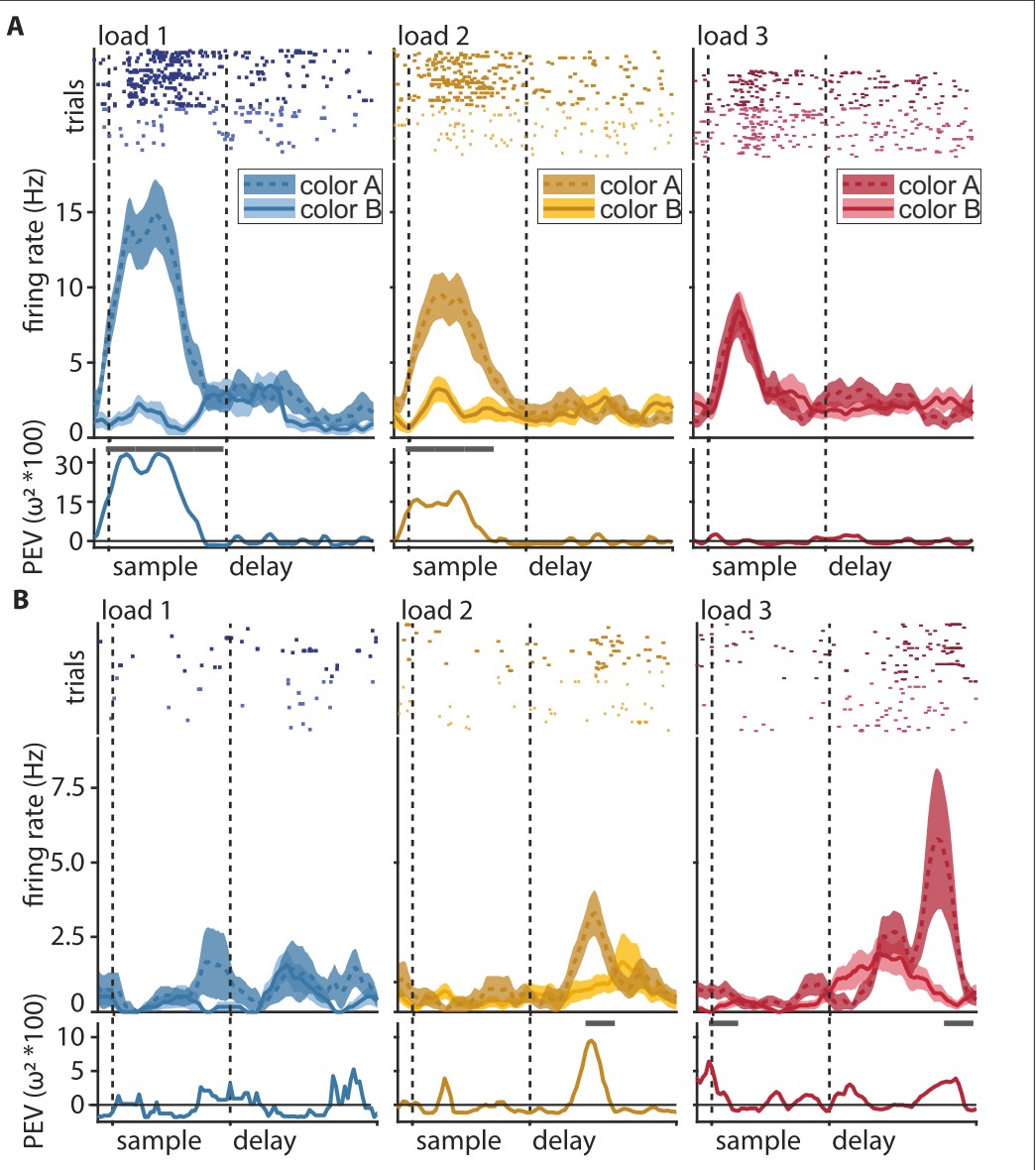

**Figure 2.** Color discrimination in the neuronal response (information, percent explained variance [PEV]) generally decreases with load, but some neurons show the opposite effect. Shown are the three ipsilateral load conditions (i.e., load increases on the same side as the neuron's favorite location). Ipsilateral loads are one (blue), two (yellow), and three (red). The labels 'color A' and 'color B' always refer to the same pair of colors at the neuron's favorite location, irrespective of the load condition. (**A**) Example of a sample neuron with color information decline at load 1 (blue), load 2 (yellow), and load 3 (red). Top: raster plot, where every dot represents a single spike during the individual trials (rows of dots); middle: peri-stimulus-time histogram (PSTH) of average firing rate (solid line for color ID 1, dashed line for color ID 2) with the standard error of the mean (shaded areas); bottom: percent explained variance of color identity (a measure of information about color) along the trial, the line at the top of the y-axis indicates significant bins. (**B**) Same as in (**A**) for an example of a delay neuron with information gain at a higher load.

The online version of this article includes the following figure supplement(s) for figure 2:

**Figure supplement 1.** Color discrimination in the neuronal response (information, percent explained variance [PEV]) decreases with load.

**Figure supplement 2.** Color discrimination in the neuronal response (information, percent explained variance [PEV]) increases with load.

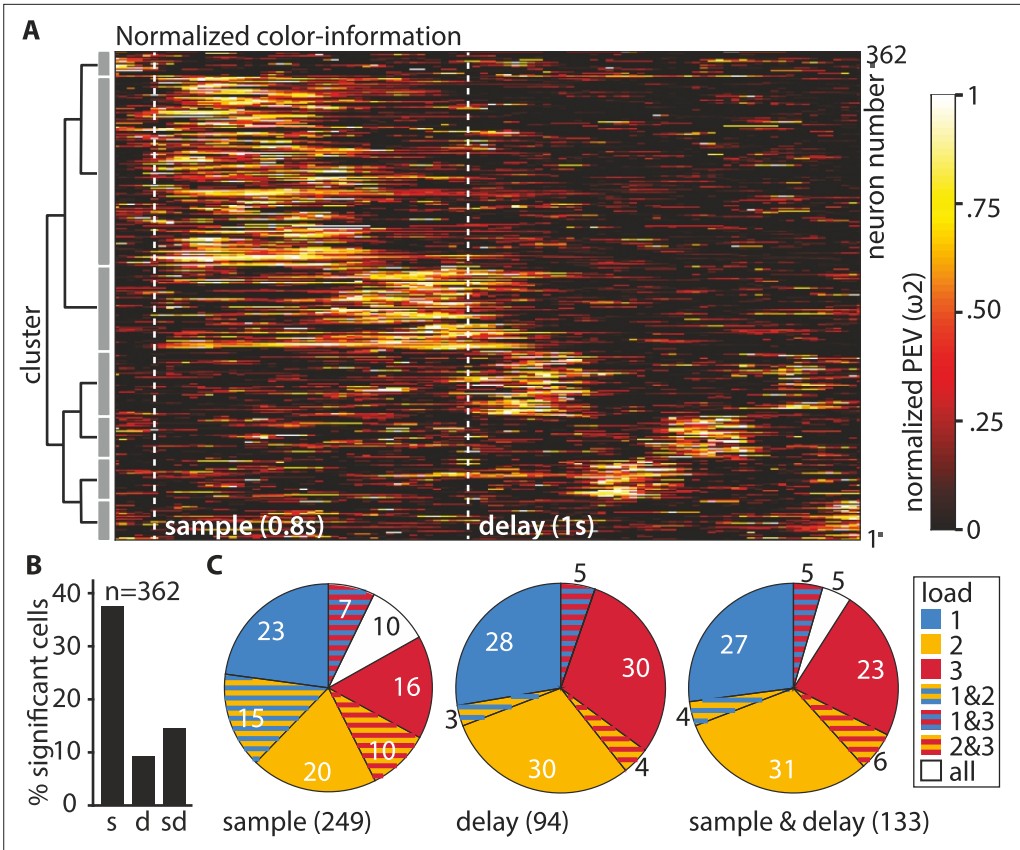

**Figure 3.** Overview of recorded neurons. (**A**) The neuronal population can be best described by seven individual clusters. (**B**) Percentages of neurons (total *n* = 362) with significant color information at load 1, during the sample and the delay. (**C**) Percentages (rounded) of significant neurons in individual load conditions for sample (*n* = 249), delay (*n* = 94), and sample and delay (*n* = 133). The pieces of the pies depicting significance at a specific load relate to the number of significant neurons in the respective phase (e.g., 36% of the 94 delay neurons (i.e., 34 neurons) are significant at load 1 (all pieces contain blue), these are the same neurons that make up the 9.36% of the total 362 neurons depicted in B).

The online version of this article includes the following figure supplement(s) for figure 3:

**Figure supplement 1.** Overview of analyses for bird 1.

**Figure supplement 2.** Overview of analyses for bird 2.

The individual neurons were then further classified into three groups depending on the phase in which they had a significant amount of information (*Figure 3B*). Overall, 37.57% (*n* = 136) of neurons were significant during the sample phase, 9.39% (*n* = 34) of neurons were significant during the memory delay, and 14.64% (*n* = 53) of neurons were significant during both the sample phase and the memory delay (all proportions of neurons were significantly higher than expected by chance [binomial test, see Materials and methods, all p < 0.001]). Refer to *Figure 2A* for an example neuron, significant at load 1 with a large differentiation in firing rate between color identities (a large PEV) and a loss of differentiation with increasing ipsilateral load. Further inspection of individual neuronal activity revealed, however, that a substantial number of neurons responded differently. Instead of losing information at higher loads, many neurons gained information (e.g., *Figure 2B*, *Figure 2—figure supplement 2*). Thus, we additionally performed the permutation testing for loads 2 and 3 to determine which neurons had significant information (see Materials and methods). We found that many of the neurons that did not have significant information at load 1 did have significant information at load 2 and load 3 (*Figure 3C*). For the memory delay, more than half of the significant neurons we detected were only significant for either load 2 or load 3, compared to only 36% of neurons that were significant at load 1 (*Figure 3C* middle). By including the higher loads in our analysis, we found a total of 249 (68.78%) sample neurons and 94 (25.97%) delay neurons. For the population analyses, we subsequently pooled

**Table 1.** Overview of significant groups.

The '+' denotes that a neuron of the respective group had a significant percent explained variance (PEV) in the respective load condition. The '-' denotes that a neuron of the respective group did not have a significant PEV in the load respective condition. The pooled groups contained only neurons with a '+' for the respective load condition.

| Load 1 | Load 2 | Load 3 | Group name | |
|--------|--------|--------|------------|---|
| + | - | - | Load 1 neurons | Group I |
| - | + | - | Load 2 neurons | Group II |
| - | - | + | Load 3 neurons | Group III |
| + | + | - | Load 1 and 2 neurons | Group IV |
| + | - | + | Load 1 and 3 neurons | Group V |
| - | + | + | Load 2 and 3 neurons | Group VI |
| + | + | + | Load 1, 2, and neurons | Group VII |
| Pooled group 1 | Pooled group 2 | Pooled group 3 | | |

all significant neurons into three groups (one per load). These pooled groups were then each subdivided into sample and delay neurons (i.e., 'sample-load1', 'delay-load1', 'sample-load2', etc., see *Table 1* in the Materials and methods for an overview).

## The neuronal population has gradually less information with increasing load

The clustering analysis indicated that the population of neurons as a whole did sustain the color information throughout the entire trial (*Figure 3A*). Plotting the information averages of each of the three 'sample populations' and the 'delay populations' over time confirmed this result (*Figure 4A*, *Figure 4—figure supplement 1*). After the onset of the stimulus array, the average information exhibited a sharp increase that peaked roughly 400 ms after stimulus onset and remained at an elevated level throughout the memory delay, until the choice array appeared. Results obtained from neurons of the lateral PFC of monkeys indicated distinct hemispheric independence of WM capacity (*Buschman et al., 2011*). This means that increasing ipsilateral load (i.e., load in the hemifield containing the target for which information is assessed) should affect neuronal processing while increasing contralateral load should not. This effect might be further emphasized in birds due to the full decussation of their optic nerve (*Husband and Shimizu, 2001*). Parallel to the behavioral results and in line with the results from monkeys, we found a strong effect of ipsilateral load on the information maintenance, as there was a sharp drop in information when the load increased from one item to two items (*Figure 4A*, blue and yellow curves). The addition of a third item only slightly decreased the maintained information further (*Figure 4A*, red curve). The load dependence was much more pronounced during the sample period than during the memory delay where the information remained at a lower elevated level. Notably, the load effect was only present for ipsilateral manipulations. If the number of items on the contralateral side was increased, the information encoded about the colors at the favorite location did not change (*Figure 4A*, right). To compare our results to the results obtained in monkeys we also applied the method of *Buschman et al., 2011* for testing the ipsilateral load effect during the sample and delay phase, by splitting each phase into an early and a late portion (first and second 400 ms of the sample, and first and second 500 ms for the delay). We did find a significant drop in information with an increase in load from one through three in the early ($F(2,537) = 18.73$, $p < 0.001$, $\omega^2 = 0.0616$) and late ($F(2,536) = 20.07$, $p < 0.001$, $\omega^2 = 0.0661$) sample period and the early ($F(2,267) = 6.88$, $p = 0.0012$, $\omega^2 = 0.0417$) and late ($F(2,267) = 3.85$, $p = 0.0225$, $\omega^2 = 0.0207$) delay period (*Figure 4B*). There was a large and significant drop between one and two items (post hoc Bonferroni corrected multiple comparisons: early and late sample $p < 0.001$, early delay $p < 0.001$, late delay $p = 0.0198$) and one and three items (post hoc Bonferroni corrected multiple comparisons: early and late sample $p < 0.001$, early delay $p = 0.019$, late delay $p > 0.05$) but no difference between loads 2 and 3 (post hoc Bonferroni corrected multiple comparisons: all $p > 0.05$). The maintenance of a significant amount

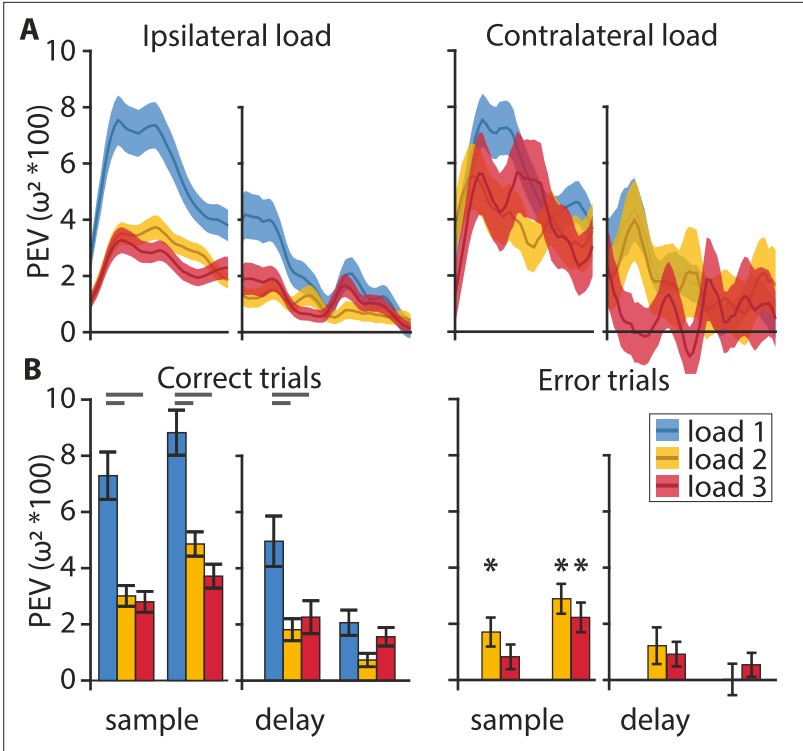

**Figure 4.** Information encoding at the population level. (**A**) Color information (percent explained variance [PEV]) decreases with an increasing ipsilateral load (i.e., on the same side as the neuron's favorite location) but not with an increasing contralateral load (i.e., on the opposite side to the neuron's favorite location). (**B**) On correct trials (left) color is represented during the early and late phase of the sample and, to a lesser degree, during the early and late delay. On error trials (right), color information can be found in the early sample phase at load 2, and in the late sample phase at loads 2 and 3 (asterisks). Analysis of load 1 error trials was omitted due to their very low abundance. Statistical comparisons of correct vs. error trial information were performed on sub-sampled correct trials. Early and late sample each 400 ms, early and late delay each 500 ms, shaded areas, and error bars indicate the standard error of the mean.

The online version of this article includes the following figure supplement(s) for figure 4:

**Figure supplement 1.** Sample population (**A**) and delay population (**B**), same as *Figure 4A* with full time axis.

**Figure supplement 2.** Same as *Figure 4B*, after applying a more stringent criterion on neuronal significance (see text).

---

of information at higher loads (even for three items, early sample $t(156)$ = 7.55, p < 0.001; late sample $t(156)$ = 8.73, p < 0.001; early delay $t(87)$ = 3.84, p < 0.001; late delay $t(87)$ = 4.73, p < 0.001) and its gradual reduction when items were added to the corresponding hemifield are indicative of a flexible resource allocation and not an all-or-nothing slot-like WM. Furthermore, if there is a flexible resource, in error trials a small but insufficient amount of resource might still be allocated to an item. Indeed, error trial analysis (applying correct trial sub-sampling, see Materials and methods) for the load 2 and 3 conditions further supported this interpretation. The amount of information in the early and late sample phase remained above zero (load 2: early, $t(186)$ = 3.25, p = 0.0014; late, $t(186)$ = 5.33, p < 0.001; load 3: $t(156)$ = 4.21, p < 0.001; *Figure 4B* asterisks), and was significantly smaller than in correct trials (load 2: late, $t(186)$ = 2.81, p = 0.0055, $d$ = 0.26; load 3: late $t(156)$ = 2.55, p = 0.0117, $d$ = 0.23). Additionally, there was no further maintenance throughout the memory delay at any load (*Figure 4B*, PEV at loads 2 and 3 in error trials delay, all non-significant). This indicates that a failure to report which color had changed at higher loads (two and three ipsilateral items) resulted from a smaller amount of information encoding during the sample phase that was not maintained throughout the delay. A possible alternative 'slot-model' explanation would be that, on error trials, the color information was completely lost after the sample phase, because it was not successfully transferred into a slot (or that a slot was not available to take on information). The graded amount of information

on correct trials is not compatible with the simple (all or none) slot model, but could fit the 'slots and averaging model' (*Zhang and Luck, 2008*).

## Higher loads produce divisive normalization-like neuronal responses

We next wanted to understand the neuronal mechanisms behind the information loss at higher WM loads. For that, we analyzed how the responses of individual sample and delay neurons changed when the load increased from one color to two colors. For the 'sample populations' and the 'delay populations', an increasing number of items reduced the amount of encoded information about the color identity (*Figure 4*). This effect was due to neurons that had a large difference of firing rates between the color A and color B at load 1 (high PEV, i.e., information about color), and reduced differentiation at load 2 (small PEV, no or little information about color, e.g., *Figure 2A*). 'Divisive-normalization-like regularization' (DNR; *Carandini and Heeger, 2011*) can explain this effect. DNR describes the computation that takes place when two stimuli are presented simultaneously. In a simplified case a neuronal response becomes normalized, analogous to vector normalization, with a normalization factor consisting of the simultaneous stimuli (*Carandini and Heeger, 2011*). Applied to our context, a consequence of DNR would be a reduced differentiation between two color identities at load 2 because differences in firing rate (for each stimulus by itself) at load 1 would be normalized at load 2 (resulting in information loss, e.g., *Figure 2A*). We, therefore, hypothesized that DNR was observable for neurons with significant information at load 1. We tested for DNR in the NCL by calculating a selectivity index (SE) and a sensory interaction index (SI) for each neuron for the sample phase and the memory delay phase (*Reynolds et al., 1999*, see Materials and methods for details). SE indicates how strongly the neuronal response is driven by a color at the favorite location of the neuron (reference) in relation to a selected probe color (when either is presented alone). SI indicates how the probe color interacts with the reference color when both are presented simultaneously. Values of both indices, SE and SI, lie between −1 and +1. The addition of a probe color influences the response to the reference color by either suppressing the firing rate of the reference color (if the reference elicits a higher firing rate than the probe, i.e., SE <0), or increasing the firing rate for the reference color (if the probe elicits a higher firing rate than the reference, i.e., SE >0). If DNR was present, this influence to suppress or enhance neuronal responses should be an even mixture at the population level, resulting in a significant regression between SE and SI with a slope of around 0.5 (*Bouchacourt and Buschman, 2019*). We compared regressions for the sample and delay phase (each as one bin, see Materials and methods for details) for two groups of neurons: information-carrying neurons (significant information at load 1), and non-informative neurons (no information at load 1 or at load 2; *Figure 5—figure supplement 1*). We found that DNR was present in both sample and delay phases (*Figure 5A*). Information-carrying sample neurons had a fitted slope of 0.47 ($R^2_{adj}$ 0.39, $F(1,838) = 547.69$, p < 0.001, CI = [0.43 0.51]) and delay neurons had a slope of 0.50 ($R^2_{adj}$ 0.34, $F(1,342) = 175.60$, p < 0.001, CI = [0.43 0.58]). As the slopes were not significantly different from 0.5, this indicates that reference and probe color had an equal influence on neuronal responses. We thus show that DNR was observable in the neuronal population, and as a consequence of this computation, neurons had generally less information about the color identity at load 2.

## Gain of information at load 2 can be explained by neuronal normalization

Some neurons showed encoding of color identity at higher loads, instead of loss of information. These neurons were abundant in both the sample phase and the delay phase (*Figure 3C*). For example, the neuron shown in *Figure 2B* did not differentiate between color identities at load 1 but did so for load 2, thus, representing a case of information gain (instead of loss) at a higher load. We wanted to understand if DNR, the mechanism that we found reduced color information at load 2, could also produce color differentiation. The 'normalization model of attention' (*Reynolds and Heeger, 2009*) incorporates divisive normalization, and can explain how attention can modulate neuronal responses. By attending a preferred (or non-preferred) second colored square in the load 2 condition the neuronal response of a neuron to the target location (i.e., to color A and to color B at the favorite location) might be altered. As a result a difference between color A and B may arise even though each color by itself elicited a similar response. In other words, the interaction between the additional color and the target color is unequal. Neurons without a color differentiation at load 1 that gained differentiation

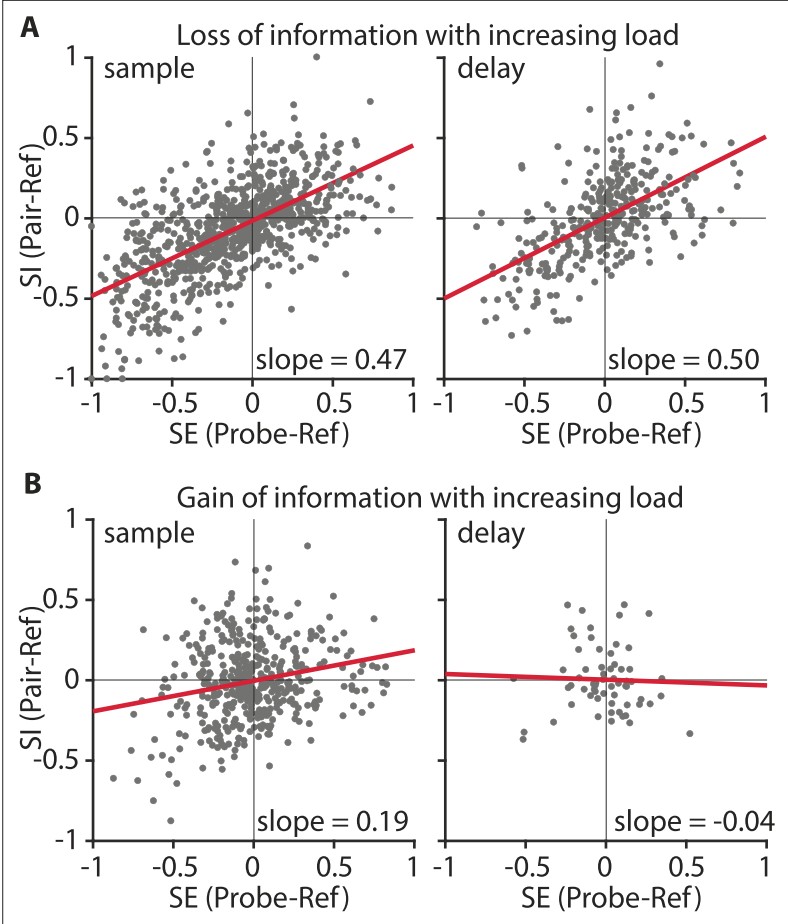

**Figure 5.** Divisive normalization-like regularization was observable for neuronal responses of neurons losing information (**A**) but not for neurons gaining color information at load 2 (**B**). Selectivity (SE) indicates how much the neuronal response is influenced by a color, relative to a second color when either is presented alone. Sensory interaction (SI) indicates how much the neuronal response is influenced by either color when both were displayed simultaneously. Slopes close to 0.5 indicate an equal influence of both colors. Slopes <0.5, or >0.5 indicate a weighted influence of a color. (**A**) Information-carrying neurons in the sample phase (as one bin; n = 105; left) and delay phase (as one bin; n = 43; right) population. (**B**) Information gaining neurons in the sample phase (as one bin; n = 56; left) and delay phase (as one bin; n = 8; right) population. The red line indicates the regression fit.

The online version of this article includes the following figure supplement(s) for figure 5:

**Figure supplement 1.** Divisive normalization-like regularization was observable for neuronal responses of neurons without significant information.

**Figure supplement 2.** Example for information gain due to unequal interaction at load 2.

at load 2 through this process (e.g., if the interaction of probe color A with reference color A is larger than the interaction of probe color A with reference color B, see *Figure 5—figure supplement 2* for an example) should have a population regression slope smaller than 0.5. We thus hypothesized that the population of neurons showing information at load 2, but not at load 1 (e.g., *Figure 2B*), would have a smaller slope than the neurons that lost information (*Figure 5A*). Sample neurons had a slope of 0.19 ($R^2_{adj}$ 0.05, $F(1,446)$ = 23.0, p < 0.001, CI = [0.11 0.27], *Figure 5B*), and delay neurons had a slope of –0.04 ($R^2_{adj}$ –0.015, $F(1,62)$ = 0.08, p = 0.78, CI = [–0.29 0.22], *Figure 5B*). Both slopes were significantly smaller than 0.5 and smaller than the slopes of the non-informative neurons (*Figure 5—figure supplement 1*). This indicates that these neurons were influenced more strongly by the reference color, and that the addition of the probe color at load 2 resulted in an unequal interaction. Therefore, DNR was also computationally responsible for a gain of information at load 2, in a specific subset of neurons.

## Discussion

### Neuronal resources of WM capacity are hemifield independent and gradually allocated

Our results confirm behavioral findings that have been discussed in detail in an earlier study (**Balakhonov and Rose, 2017**). In brief, we found that the WM capacity of crows is limited to about four items, and that the two visual hemifields are largely independent (i.e., the number of items on one side does not affect change detection performance on the other side). Within each hemifield, performance dropped gradually with the addition of a second and third item but remained above chance. Fittingly, on the neuronal level, we found a markedly reduced amount of color information when the number of colored squares was increased from one to two (roughly 50% reduction in correct trials). This suggests that WM could be conceptualized as a continuous resource that has to be divided between the two items (**Bays and Husain, 2008**; **van den Berg et al., 2012**; **Wilken and Ma, 2004**), rather than two 'simple' slots that would each have the same amount of information irrespective of the memory load. This is also consistent with results of human neuroimaging that report decreased signal amplitude and precision with increasing memory load (**Emrich et al., 2013**; **Sprague et al., 2014**). In contrast, the hemispheric independence we observed would fit a slot-like model, in which the hemispheres as a whole act like discrete slots. A more nuanced version of the slot model ('slots and averaging'; **Zhang and Luck, 2008**) could also account for graded amounts of information within a limited number of slots (**Fukuda et al., 2010**; **Zhang and Luck, 2008**), as we found here. The mix of discrete and independent hemispheres with a graded allocation of information between items that we found is comparable to results by **Buschman et al., 2011**, observed in monkey PFC. On the neuronal level, recurrent connections between neurons within a hemisphere may reduce item differentiation when multiple items are present simultaneously, creating capacity limitations within the hemisphere (**Matsushima and Tanaka, 2014**). A lack of interhemispheric recurrent connections would make processing in the other hemisphere independent. Like in monkeys, WM capacity in crows may therefore result from neuronal activity patterns governed by multiple individual items. We probed the WM capacity of crows using colored squares, based on the task design of **Buschman et al., 2011**. Using the identical task allowed us to directly compare our neuronal results of WM capacity from NCL to results from PFC of monkeys. Task similarity is very important for such cross species comparisons as even small changes in task parameters may introduce substantial differences in neuronal responses, leading to potentially different conclusions. In a task similar to the one used here, **Lara and Wallis, 2014**, have found that neurons in the PFC of monkeys encoded nearly no information about color, but instead about location. In their task monkeys had to memorize the color of squares at two locations on a screen, and were again confronted with a colored square at one of the two locations after a delay. The monkeys then had to indicate if the color at that location had changed. **Lara and Wallis, 2014**, discuss the absence of color information in the neurons they recorded in relation to the task of **Buschman et al., 2011**, who like us, did find color information. In brief, the exact task design may determine the neuronal encoding of task relevant information (**Lara and Wallis, 2014**). Similar to the complex contribution of PFC neurons to WM, neurons of NCL can also encode a wide range of very different task relevant aspects, like color (this study), spatial locations (**Rinnert et al., 2019**; **Veit et al., 2017**), and more abstract items like rules (**Veit and Nieder, 2013**) and numerosities (**Ditz and Nieder, 2015**).

### Attentional processes guide WM allocation and maintenance

One way to circumvent WM failure when item load increases is to allocate attention. Our results suggest that attention may play an important role in crow WM. Capacity limitation became apparent during encoding, as the amount of information at the end of the sample period was affected by the stimulus load. Adding a second and third item to the ipsilateral stimulus array reduced the amount of color information encoded by NCL neurons that carried over into the memory delay. Furthermore, neuronal activity in trials in which the birds made an incorrect response showed only weak encoding during the sample phase without information maintenance during the memory delay. This fits studies of human WM that have shown attentive filtering during encoding of stimuli influencing WM capacity (**Bays and Husain, 2008**; **Vogel et al., 2005**; **Vogel and Machizawa, 2004**), and neuronal correlates of this have been reported for monkeys as well (**Buschman et al., 2011**). Beyond the domain of sensory signals, attention and WM may be directly linked. Neuronal correlates of WM and attention overlap in PFC neurons, for example, **Lebedev et al., 2004**, found that a substantial amount of PFC neurons

encode either an attentional signal, or a memory signal, and some (hybrid) neurons do both. A purely mnemonic function of PFC thereby seems unlikely. Indeed, very recently, *Panichello and Buschman, 2021*, have reported that at the population level neurons of PFC encode 'both the selection of items from working memory and attention to sensory inputs' (p. 2), rather than just memory content. The independence of hemifields that we observed on the behavioral level (this study and *Balakhonov and Rose, 2017*) and found in the neuronal responses could also be related to attention. Adding stimuli in the contralateral hemifield affected neither performance nor information maintained by NCL neurons, whereas additional ipsilateral stimuli strongly reduced both. This fits the influence of attention on WM and hemifield independence, which is consistently accentuated in studies in which attention had to be divided between the two hemifields (*Alvarez and Cavanagh, 2005*; *Buschman et al., 2011*; *Cavanagh and Alvarez, 2005*; *Delvenne, 2005*; *Delvenne et al., 2011*). Finally, the DNR computation may explain the responses of the neurons that gained information at load 2 through attentional processes predicted by the 'normalization model of attention' (*Reynolds and Heeger, 2009*). This may appear counter-intuitive and contradictory, considering that the same process is also responsible for the loss of information. However, when attention is overtly directed to a specific (preferred or non-preferred) item within the receptive field of a neuron, the DNR computation shifts its weighting of the normalized response toward the response of the attended item (*Reynolds et al., 1999*; *Reynolds and Heeger, 2009*). This weighted normalization can produce a difference in the neuronal response to both color identities at load 2, even if the neuronal response was non-informative at load 1. At the population level we were able to observe such an effect as the reduced slope of the selectivity/ interaction fit. Thus, an attentive process might have enhanced information in WM at higher loads. It is important to clarify that, as we did not use any form of attentional cueing in our study, we cannot explicitly test for such an attention effect. However, we do know that the animals participating in this study can use attentional cues to enhance their WM (*Fongaro and Rose, 2020*). The attention cues used by *Fongaro and Rose, 2020*, positively affected not only encoding but also the maintenance and retrieval of the information held in WM, comparable to results from monkeys and humans (*Brady and Hampton, 2018*; *Souza and Oberauer, 2016*). We, therefore, want to emphasize that our data is in line with the interpretation that the birds possibly attended a load 2 stimulus array differently than a load 1 stimulus array in order to enhance their performance in trials with higher loads.

## Modern models of mammalian WM capacity are applicable to crows

Our neuronal recordings offer a mechanistic explanation for the behavioral effects, as we found clear evidence of DNR governing the neuronal responses tied to WM capacity that is in accordance with mammalian models of WM capacity (*Bouchacourt and Buschman, 2019*; *Lundqvist et al., 2016*; *Lundqvist et al., 2011*). The loss of information about color identity (i.e., neuronal response differentiation between colors) can be accounted for by DNR when an item is added to a neuron's receptive field. The normalization of neuronal firing rate diminishes the differentiation between color identities. As such it is analogous to neurophysiological responses from visual areas (*Carandini et al., 1997*; *Reynolds et al., 1999*) and to the PFC during spatial WM (*Matsushima and Tanaka, 2014*). The WM model of *Bouchacourt and Buschman, 2019*, is based solely on data from monkey electrophysiology, and thus implicitly tied to the layered columns of the neocortex. The results we report here show that the model also fits the neurophysiology of WM in crows. However, the picture is incomplete since important aspects of monkeys' WM are still not investigated in crows. Oscillations of local field potentials are relevant for how information enters WM and how it is maintained (*Miller et al., 2018*), and have been tied to normalization and competition between items in WM (*Lundqvist et al., 2018b*). Thus, the oscillatory interplay of the layers and different regions of the mammalian neocortex are important fields of research to further our understanding of WM. Such aspects are so far completely unknown in crows and their non-layered associative areas. This encourages further investigation into the neuronal circuits of WM in birds. There is also ongoing debate about the role of sustained activity during delay periods and how it relates to WM (*Constantinidis et al., 2018*; *Lundqvist et al., 2018a*; *Miller et al., 2018*). We cannot report of any neuron that showed persistent activity comparable to those reported by classical WM studies in PFC (e.g., *Funahashi et al., 1989*; *Fuster and Alexander, 1971*), or in NCL (*Diekamp et al., 2002b*; *Veit et al., 2014*; *Veit and Nieder, 2013*). This may be reconcilable with some other contemporary models of WM. One major type of those models implements 'synfire chains', where individual neurons fire sequentially (and transiently) to bridge temporal

gaps and maintain task relevant contents (*Rajan et al., 2016*). This has, for example, been reported to be the case in posterior parietal cortex of mice performing a T-maze task that required WM for cued spatial locations to be maintained (*Harvey et al., 2012*). The transient activity of neurons that we report (*Figures 2 and 3A*) might fit into such models. However, our results can only be compared very cautiously to this (since even small changes in task design significantly alter neuronal responses, e.g., *Lara and Wallis, 2014*). Therefore, while we cannot, yet, fully equate crow and monkey WM, our results raise two important questions about how WM is implemented on the level of neuronal networks that have implications for our comparative view of crow WM. The first regards the neuronal computations underlying WM. Is there a common canonical computation governing WM, or are there different solutions based on different neuronal architectures? Recent work has shown that the sensory areas of mammals and birds show remarkably similar circuit organization (*Stacho et al., 2020*). However, higher-order associative areas involved in WM, like the LPFC in mammals and the NCL in birds, have distinctly different architectures (*Stacho et al., 2020*). The fact that differently organized areas like LPFC and NCL produce strikingly similar physiological responses points to shared computational principles. Modeling work already suggests that the competing WM capacity models can be accommodated into a unifying framework based on theoretical neuronal information sampling, where stochastic information sampling (assumed for continuous resource models) can account for item limitations better than fixed information sampling (assumed by the slots and averaging models) (*Schneegans et al., 2020*). Similarly, DNR is already considered to be a general, canonical computation of the nervous system, present in evolutionarily distant phyla, for example, fruit flies and monkeys (*Carandini and Heeger, 2011*). The second question regards the tradeoff between WM flexibility and capacity (*Bouchacourt and Buschman, 2019*). Is the WM of a crow as flexible as that of a monkey? Our results show that the computations by individual neurons that result in WM capacity limitations are virtually the same in crows and monkeys, highlighting a further aspect of WM that is similar between these animal groups (*Nieder, 2017*). Ultimately, our results were in line with different modern models of WM that implement DNR to explain capacity (*Bouchacourt and Buschman, 2019*; *Lundqvist et al., 2016*; *Lundqvist et al., 2011*; *Schneegans et al., 2020*). However, the data we presented cannot carry a definitive conclusion about which of the different models fits best. For example, a tradeoff between flexibility and capacity (*Bouchacourt and Buschman, 2019*) might be present, but further investigation into the models' predictions is required. We do, however, show that mammalian models of WM are in line with WM in birds, which implies that fundamental aspects of WM are shared between these animal groups.

## Conclusion

Together, all these facets of crow WM capacity suggest that the different intricate neuronal architectures that carry out the computations in monkeys and crows have likely been shaped by convergent evolution – into systems that yield similar cognitive performances. The systems may share the same basic mechanisms and thus limitations. Further investigation into the oscillatory dynamics of WM in the avian brain may elucidate if birds also share the prominent limitation of a tradeoff between flexibility and capacity.

# Materials and methods
## Subjects

Two hand-raised carrion crows (*C. corone*) of 2 years of age served as subjects in this study. The birds were housed in spacious aviaries in social groups. During the experimental procedures, the animals were held on a controlled food protocol with ad libitum access to water and grit. All experimental procedures and housing conditions were carried out in accordance with the National Institutes of Health *Guide for Care and Use of Laboratory Animals* and were authorized by the national authority (*LANUV*).

## Experimental setup

We used operant training chambers (50 × 50.5 × 77.5 cm³, width × depth × height) equipped with an acoustic pulse touchscreen (22", ELO 2200L APR, Elo Touch Solutions Inc, Milpitas, CA) and an infrared camera (Sygonix, Nürnberg, Germany) for remote monitoring. The birds sat on a wooden

perch so that the distance between the bird's eye and the touchscreen was 8 cm. Food pellets were delivered as a reward via a custom-made automatic feeder (plans available at http://www.jonasrose.net/). The position of the animal's head was tracked online during the experiment by two open-source computer vision cameras ('Pixy', CMUcam5, Charmed Labs, Austin, TX) that reported the location and angle between two LEDs. For tracking, we surgically implanted a lightweight head-post and used a lightweight 3D-printed mount with LEDs that was removed after each experimental session. The system reported the head location at a frame rate of 50 Hz and data was smoothed by integrating over two frames in MATLAB using custom programs on a control PC. All experiments were controlled by custom programs in MATLAB using the Biopsychology (*Rose et al., 2008*) and Psychophysics tool-boxes (*Brainard, 1997*). Digital input and output of the control PC were handled by a microcontroller (ODROID C1, Hardkernel co. Ltd, Anyang, South Korea) connected through a gigabit network running custom software (available at http://www.jonasrose.net/).

## Behavioral protocol

The behavioral protocol was identical to the one described in *Balakhonov and Rose, 2017*. We trained the birds to perform a delayed change localization task that had previously been used to test the performance under different WM loads in primates (*Buschman et al., 2011*). Each trial started after a 2 s inter-trial-interval, with the presentation of a red dot centered on the touchscreen (for a maximum of 40 s). The animals initiated the trial by centering their head in front of the red dot for 160 ms. This caused the red dot to disappear and a stimulus array of two to five colored squares to appear (*Figure 1A*, 'sample'). The colored squares were presented for a period of 800 ms, during which the animals had to hold their head still and center their gaze on the screen ('hold gaze', no more than 2 cm horizontal or vertical displacement, and no more than 20 degrees horizontal or vertical rotation). Failure to hold the head in this position resulted in an aborted trial. This sample phase was followed by a memory delay of 1000 ms after which the stimulus array reappeared with one color exchanged. The animal had to indicate the location of the color change by pecking the respective square. Correct responses were rewarded probabilistically (BEO special pellets, in 55% of correct trials, additional 2 s illumination of the food receptacle in 100% of correct trials). Incorrect responses to colors that had not changed or a failure to respond within 4 s resulted in a brief screen flash and a 10 s timeout. The stimuli were presented at six fixed locations on the screen (1–6, *Figure 1A*). In each session, one pair of colors was assigned to each of the six locations. Each location had its own distinct pair. These pairs were randomly chosen from a pool of 14 colors (two color combinations were excluded since the animals did not discriminate them equally well during a pre-training). Let us consider *Figure 1A* as an example. The color change occurs in the middle-left where turquois (T) is presented during the sample and orange (O) during the choice. In this particular session the middle-left could thus show either of the following colors during the sample and choice: T-O (shown in *Figure 1A*); O-T; O-O; T-T; None-None. On the next session a new random pair of colors was displayed at this location.

For identification and analysis an arbitrary label was assigned to each of the randomly drawn colors at the start of each session (i.e., left middle location: 'color A', or 'color B'; left top location: 'color A', or 'color B'; etc.). These indices do not refer to the order of presentation of the colors at any time, but were held constant for neuronal analysis. The order of presentation of colors within a pair, the target location (where the color change occurred), and the number of stimuli in the array (two to five) were randomized and balanced across trials so that each condition had an equal likelihood to appear. The order of presentation of colors within a pair, the target location (where the color change occurred), and the number of stimuli in the array (two to five) were randomized and balanced across trials so that each condition had an equal likelihood to appear. The color squares had a width of 10 degrees of visual angle (DVA) and were placed on the horizontal meridian of the screen and at 45.8 DVA above or below the meridian at a distance of 54 and 55.4 DVA from the center. This arrangement in combination with the head tracking ensured that all stimuli appeared outside of the binocular visual field of crows (37.6 DVA; *Troscianko et al., 2012*).

## Surgery

Both animals were chronically implanted with a lightweight head-post to attach a small LED holder during the experiments. Before surgery, animals were deeply anesthetized with ketamine (50 mg/kg) and xylazine (5 mg/kg). Once deeply anesthetized, animals were placed in a stereotaxic frame.

After attaching the small head-post with dental acrylic, a microdrive with a multi-channel microelectrode was stereotactically implanted at the craniotomy (Neuronexus Technologies Inc, Ann Arbor MI, DDrive). The electrode was positioned in NCL (AP 5.0, ML 13.0) of the left hemisphere (coordinates for the region based on histological studies on the localization of NCL in crows; *Veit and Nieder, 2013*). After the surgery, the crows received analgesics.

## Electrophysiological recordings

Extracellular single neuron recordings were performed using chronically implanted multi-channel microelectrodes. The distance between recording sites was 50 μm. The signal was amplified, filtered, and digitized using Intan RHD2000 headstages and a USB-Interface board (Intan Technologies LLC, Los Angeles, CA). The system also recorded digital event codes that were sent from the behavioral control PC using a custom IO device (details available at http://www.jonasrose.net/). Before each recording session, the electrodes were advanced manually using the microdrive. Recordings were started 20 min after the advancement, and each recording site was manually checked for neuronal signals. The signals were recorded at a sampling rate of 30 kHz and filtered with a band-pass filter at recording (0.5–7.5 kHz). The recorded neuronal signals were not pre-selected for task involvement. We performed spike sorting using the semi-automatic Klusta-suite software (*Rossant et al., 2016*), which uses the high electrode count and their close spacing to isolate signals of single neurons. For spike sorting, we filtered with a high pass of 500 Hz and a low pass of 7125 Hz. The software utilizes the spatial distribution of the recorded signal along the different recording sites to untangle overlapping signals and separate signals with similar waveforms but different recording depths.

## Data analysis

All statistical analyses were performed in MATLAB (2018b, Mathworks Inc) using commercially available toolboxes (Curve Fitting Toolbox Version 3.5.3, Statistics and Machine Learning Toolbox Version 10.2) and custom code. For all statistical tests, we assumed a significance level of $\alpha = 0.05$, unless stated otherwise. Trials were classified as error trials if the bird chose a location where no change of colors had appeared. Trials in which the bird did not choose any location or failed to maintain head fixation were not analyzed. All correct trials were included in the analysis of neural data. Depending on the analysis we refer to different 'load conditions' relative to referential sides of the screen which are either ipsilateral (same side) or contralateral (opposite side), each with a possible load between one and three items. For the behavioral analyses the terms ipsilateral and contralateral refer to the location of the change that had to be detected. For the neurophysiological analyses the terms ipsilateral and contralateral refer to the respective neuron's favorite location (described in the following section).

Because there were only very few error trials in the load one condition, we performed error trial analysis only for the load 2 and load 3 conditions. The behavioral data were analyzed as described in our previous study (*Balakhonov and Rose, 2017*), estimating the WM capacity $K$ for each load by *Equation 1*.

$$K = n * p \tag{1}$$

where $p$ is the percentage correct and $n$ is the number of items in WM. This estimate has been applied to similar primate data and in studies with humans (*Johnson et al., 2013*; *Kornblith et al., 2016*).

## Information about color Identity

Based on a one-way ANOVA of color identity at a given location, we calculated a PEV statistic to measure the effect size of neuronal modulation. Its main parameter $\omega^2$ is a measurement for the percentage to which the tested factor can explain the variance of the data, and it is calculated from the sum of squares of the effect ($SS_{effect}$) and the mean squares of the within-group (error) variance ($MS_{error}$) (*Equation 2*).

$$\omega^2 = \frac{SS_{effect} - df * MS_{error}}{SS_{total} + MS_{error}} \tag{2}$$

For each neuron, we determined a 'favorite location', which was defined as the location with the highest cumulative PEV, of the three possible locations on the right half of the screen, that is, opposite to the implanted hemisphere, across four non-overlapping bins during the sample phase (bin size 200 ms, advanced in steps of 200 ms, from start till the end of the sample phase). The significance of calculated effect size values was determined by a permutation test. We ran the permutation to calculate the likelihood of getting an explained variance value bigger than the one calculated from the actual distribution of the data by randomly permuting the color identity labels and calculating the PEV 1000 times. The test thereby does not assume any distribution of the data and returns an unbiased estimate of the likelihood of generating an effect size within the data randomly. The measured value of explained variance from the actual dataset was assumed to be significant if the likelihood of randomly generating a bigger value was below 5%. We chose to not correct for the multiple comparisons at this level, as we reasoned that if we were to only include those neurons that had the most information at the individual loads (i.e., those with the highest PEV values that are significant even under very stringent statistical criteria) we would have artificially inflated the amount of information present at each load. By including neurons that encoded less information too (i.e., at the uncorrected p-value) our analysis population was more resistant to such outlier effects. We additionally performed our analyses using a more stringent statistical criterion (significance of two consecutive non-overlapping bins) and found the same results (*Figure 4—figure supplement 2*). We tested the proportions of significant neurons we found for the different trial phases by performing a binomial test, assuming a significance level $\alpha$ = 0.05 (*Equation 3*).

$$P\left(X=i\right) = B\left(p_0, n\right) = \left(nk\right) p_0^i \left(1 - p_0\right)^{n-i} \tag{3}$$

Calculating the probability p, of finding *X* significant neurons, given a total amount of *i* (362) neurons, and a probability $p_0$ of 5% finding a significance by chance.

## Population analyses

We considered neuronal significance (i.e., significant PEV as determined above) for each load independently. This means, we tested if the PEV of a neuron was significant three times with the permutation method described above: once for each of the three load conditions. Therefore, we can report seven groups of significance (*Table 1*, *Figure 3C*). Subsequently, we created three pooled groups (*Table 1*) from all neurons with a significant PEV at each individual load. We used these pooled groups for the population analyses (*Figures 4 and 5*). Neurons of these pooled groups, with a significant PEV during the sample phase were assigned to the 'sample population', and neurons with a significant amount of information during the memory-delay phase were assigned to the 'delay population' (significance criterion: one significant 200 ms bin, at $\alpha$ = 0.05, see above for our reasoning not to correct for multiple comparison at this point). Thus, neurons with significant PEV during both the sample and delay phase were included in both subpopulations. We corrected for the unequal amount of correct and error trials when comparing information about color (PEV) between the trial conditions, by sub-sampling correct trials with the number of error trials 1000 times for each neuron. The resulting PEV values of correct trials were then averaged for each neuron, this population of averaged PEV values was then statistically tested against the PEV values of error trials (of the same neurons) using a dependent t-test.

## Divisive normalization-like regularization

We tested for the presence of divisive normalization using the method of *Reynolds et al., 1999*. Three conditions were considered: (1) neuronal response to stimulus A, (2) neuronal response to stimulus B, and (3) neuronal response to the simultaneity of stimuli A and B. As we wanted to relate this to the information about color identity, we selected subsets of the favorite location and the additional two ipsilateral locations. To test how the neurons altered their response when multiple stimuli were presented simultaneously, we calculated the color selectivity index (SE) and the sensory interaction index (SI) of each neuron. $SE_i$ was calculated by subtracting the normalized firing rate for the chosen reference color i ($REF_i$) at the neuron's favorite location, from a second color j ($PROBE_j$) at a different location (ipsilateral to the favorite location, *Equation 4*).

$$SE_i = PROBE_j - REF_i \qquad (4)$$

The resulting selectivity index lies between –1 (completely selective for the reference color) and 1 (completely selective for the probe color). SI was calculated (*Equation 5*) by subtracting the normalized firing rate for $REF_i$ from the normalized firing rate of the combination of $REF_i$ and $PROBE_j$ ($PAIR_{i,j}$).

$$SI_{i,j} = PAIR_{i,j} - REF_i \qquad (5)$$

This interaction index also lies between –1 (full suppression of reference stimulus by the probe stimulus) and 1 (full enhancement of the reference stimulus by the probe stimulus). As each of the three locations had two possible colors, we calculated eight SE and SI indices per neuron and performed a linear regression for all indices. This is required as each stimulus combination is informative about the normalization. The effects of divisive normalization were compared between the sample and the delay phase. Therefore, SE and SI indices were calculated across the entire sample (800 ms) and memory delay (1000 ms) phase. Neurons with significant information were accordingly identified over the entire sample and delay as one bin, using the permutation test described in the section '*information about color identity*'. We considered the entire sample and delay phase because we wanted to analyze the population response as a whole, irrespective of highly diverse response profiles of individual neurons.

## Hierarchical clustering

To visualize the different groups of neurons that encoded and maintained information about the color identity during different phases of the trial, we performed a hierarchical clustering analysis in MATLAB on the normalized PEV values of individual neurons throughout the trial. We used a (1 − correlation) distance metric and an average distance linkage function for a maximum of seven clusters. The maximum number of clusters was first determined by calculating the clustering for different amounts of clusters (1–10) and subsequently calculating the within-cluster sum-of-squares. This resulted in a graph that allowed us to visually inspect the tradeoff between cluster number and fit improvement, from which we estimated the inflection point (elbow method). A cluster number of seven presented the best tradeoff that allowed visualization of the different groups at an acceptable clustering success. We then ordered the neuron clusters to minimize the average distance between the clusters in the dendrogram.

## Acknowledgements

We would like to thank Mikael Lundqvist for helpful comments on an earlier version of the manuscript.

## Additional information

### Funding

| Funder | Grant reference number | Author |
|---|---|---|
| Volkswagen Foundation | Freigeist Fellowship 93299 | Jonas Rose |
| Deutsche Forschungsgemeinschaft | Project B13 of the collaborative research center 874 (122679504) | Jonas Rose |

The funders had no role in study design, data collection and interpretation, or the decision to submit the work for publication.

### Author contributions

Lukas Alexander Hahn, Data curation, Formal analysis, Investigation, Methodology, Software, Visualization, Writing - original draft, Writing - review and editing; Dmitry Balakhonov, Conceptualization, Data curation, Methodology; Erica Fongaro, Data curation; Andreas Nieder, Project administration, Resources, Writing - review and editing; Jonas Rose, Conceptualization, Funding acquisition, Methodology, Project administration, Resources, Supervision, Visualization, Writing - review and editing

## Author ORCIDs
Lukas Alexander Hahn  http://orcid.org/0000-0002-0491-7954
Andreas Nieder  http://orcid.org/0000-0001-6381-0375
Jonas Rose  http://orcid.org/0000-0003-1745-727X

## Ethics
All experimental procedures and housing conditions were carried out in accordance with the National Institutes of Health Guide for Care and Use of Laboratory Animals and were authorized by the national authority (LANUV protocol no. 84-02.04.2017.A001).

## Decision letter and Author response
Decision letter https://doi.org/10.7554/eLife.72783.sa1
Author response https://doi.org/10.7554/eLife.72783.sa2

---

## Additional files

### Supplementary files
- Transparent reporting form
- Source data 1. Reported statistical results and numerical values.

### Data availability
All details of statistics reported in the manuscript is provided as a supporting file. Source data files of all figures are publicly available via dryad https://doi.org/10.5061/dryad.0k6djhb1q.

The following dataset was generated:

| Author(s) | Year | Dataset title | Dataset URL | Database and Identifier |
|---|---|---|---|---|
| Hahn LA, Balakhonov D, Rose J | 2021 | Working memory capacity of crows and monkeys arises from similar neuronal computations | https://doi.org/10.5061/dryad.0k6djhb1q | Dryad Digital Repository, 10.5061/dryad.0k6djhb1q |

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
