## [Editor Report]

In this study, Hahn et al., taught crows to perform a multi-item working memory task designed to mimic traditional monkey tasks. Using a combination of behavior and electrophysiology, the authors convincingly show that the neural mechanisms that limit working memory capacity in primates also limit working memory capacity in crows. Such cross-species comparisons are fundamental to understanding the computational constraints that are placed on cognition and the brain.

---

## [Decision Letter]

**Decision letter after peer review:**

Thank you for submitting your article "Working memory capacity of crows and monkeys arises from similar neuronal computations" for consideration by *eLife*. Your article has been reviewed by 3 peer reviewers, one of whom is a member of our Board of Reviewing Editors, and the evaluation has been overseen by Michael Frank as the Senior Editor. The following individual involved in review of your submission has agreed to reveal their identity: Mike Colombo (Reviewer #3).

Essential revisions:

The reviewers were each positive about the research and the paper. In reviews and discussion, a number of issues were raised which the authors should address in their revised manuscript. Below are the most important points, which are considered essential revisions. The individual reviewer comments with additional detail are appended to this list for the authors' reference.

1. Load-dependent increases in color tuning seem counter to many predictions, and the authors should clarify how the DNR model can explain an increase in information as the number of stimuli increase. The suggestion appears to be that, if neurons carry more information for load 2 trials compared to load 1 trials, then this reflects an interaction between the two items that increases the information signaled, but how exactly this might occur isn't clear. There was also confusion about how non-linear effects on neural responses, which are common in the literature, relate to the DNR model. The reviewers felt that a slope near 0 is not sufficient evidence to argue conclusively that the results can be explained by DNR. It would be helpful to consider other evidence in the data and/or other normalization mechanisms, and provide a specific example neuron that shows an increase in information, along with its response to the sample, reference, and paired stimuli.

2. It should be more clearly stated when working memory loads refer to the number of ipsilateral items on the display, particularly in the section on single unit analyses and in Figure 1.

3. More clarity on the task design, terminology, and figures is needed. Specifically, what it means that color pairs are "fixed" to some of the locations, and how colors are designated "color 1" and "color 2" and whether these designations are constant across loads 1 – 3 (Figure 2) should be clarified. Perhaps an example would help here. In addition, more information on the perceptual similarity (to crows) of the colors used, and how this did or did not impact performance is important. In Figures 1 and 2, the labels "hold gaze" and "sample" appear to be used interchangeably, leading to some confusion, and consistent terms would be helpful.

4. A major strength of the paper is cross-species comparisons, and the reviewers suggest that the discussion of these comparisons be expanded beyond monkey neurophysiology. For instance, how do the current findings relate to sequential activity commonly reported in non-primate mammals (versus the more stable 'attractor' dynamics in primate models of working memory)? How might this work relate to behavior and neuroimaging in humans that is suggestive of DNR?

5. The statistical corrections for multiple comparisons across time and/or groups should be clarified, particularly in the single unit analyses.

6. How do the authors reconcile the fact that the number of single neurons tuned to color tends to increase at higher loads, but there is a loss of color information at the population level?

7. Reviewers requested more detail on whether any neurons in the population sustained information across the delay, since it is relevant to the ongoing debate about sustained vs. transient working memory representations in non-human primates.

8. Neurons in Figure 3 seem to respond during the delay period in time windows of ~250 ms. Is this intrinsic to the neural response or is it related to the 200 ms smoothing window that was used? In other words, would a similar pattern be observed if smaller smoothing windows were used?

9. The authors argue that the existence of information on error trials, even on high load trials, suggests that memory is not an 'all-or-none' phenomena, and this is inconsistent with the slot models of working memory. However, it seems like this was only true for the sample period, and there was no information during the memory delay (Figure 3B). Could this be interpreted as an inability for a sample stimulus to be sustained during memory – in other words, that it didn't make it into a 'slot'?

10. While the results are conceptually consistent with classic effects in monkey lateral PFC, Lara and Wallis (Nat Neurosci, 2014) reported a near absence of color tuning in a very similar color change detection task, and instead found predominantly spatial attention signals. Can the authors discuss this discrepancy?

11. Interference between memory representation is mentioned as a source of information loss, and a 2-sec ITI likely generates a considerable amount of proactive interference. Thus is it possible that the neural outcomes are being driven by high levels of proactive interference generated by the current design?

*Reviewer #1 (Recommendations for the authors):*

1. While the results are conceptually consistent with classic effects in monkey lateral PFC, Lara and Wallis (Nat Neurosci, 2014) reported a near absence of color tuning in a very similar color change detection task, and instead found predominantly spatial attention signals. Can the authors discuss this discrepancy?

2. Is there much known about color vision in crows? Specifically, are there any subsets of colors used in this study that could have appeared more perceptually similar/different to the crows, and does that affect performance?

3. It wasn't immediately obvious whether the single unit analyses that looked at effects of working memory load were based on ipsilateral, or total load. Based on the behavioral results, it seems that it should be ipsilateral only, but can the authors clarify this?

4. How do the authors reconcile the fact that the number of single neurons tuned to color tends to increase at higher loads, but there is a loss of color information at the population level?

5. Load-dependent increases in color tuning also seem counter to many predictions. Are there any similar results in the NHP literature? Also, I wasn't able to follow how divisive normalization results in gain of information with increasing load. Can the authors elaborate on this?

6. Is it predicted that neurons without clear color tuning also exhibited load effects consistent with divisive normalization (Supp Figure 7)? If not, how is this result interpreted?

*Reviewer #2 (Recommendations for the authors):*

General Comments

1. I think my biggest concern is regarding the argument for divisive normalization like regularization (DNR), particularly for neurons that have increased information for a memory load of 2 items. The authors seem to suggest that if neurons carry more information for load 2 trials compared to load 1 trials, then this reflects an interaction between the two items such that information increases. First, how exactly the authors are envisioning this isn't clear to me. It would be helpful to provide a specific example neuron that shows an increase in information, along with its response to the sample, reference, and pair stimuli.

That being said, if I understand correctly, the authors seem to imply there is a non-linear effect on the neural response of some neurons when two stimuli are presented. This seems consistent with previous work (e.g., non-linear responses seen by Fusi). However, this doesn't seem consistent with the DNR model. The DNR model makes a fairly clear prediction that the response to two stimuli will be a (weighted) average of the response to each stimulus alone. Given this, it seems to me that the DNR model doesn't have a mechanism for explaining an increase in information as the number of stimuli increase. Typically, in the DNR framework, a slope close to zero is taken as evidence that the response to the Pair is equal to the Ref (e.g., when attention is shifted to the reference). That seems unlikely to be the case here, but highlights how a slope near 0 is not sufficient to argue that the results can be explained by a DNR effect.

Perhaps I am missing something, in which case I would suggest the authors make this point more clearly in the current manuscript. Or, if the classic DNR model can't predict the increase in information for two-item preferring neurons, then I think the current results could argue for other normalization mechanisms (or a combination of mechanisms) that could be acting in the brain.

2. It wasn't clear to me what statistics were corrected for multiple comparisons across time and/or groups. To measure sensory/memory information in individual neurons, the authors used a percent explained variance measure to test if the neuron responded differently to two different colors. As far as I can tell, this test was performed for each of six different stimulus locations and at multiple points in time (at least four). It isn't clear from the text whether the authors corrected for these multiple comparisons when determining whether a neuron was significant.

3. As noted above, the current work is consistent with behavior and neuroimaging in humans. In particular, there has been evidence from human neuroimaging work arguing for divisive normalization in working memory (e.g., Sprague et al., 2014) that seems relevant and worth citing.

4. On line 310 the authors state that "most neurons did not sustain information about color (…) throughout the entire sample or memory delay". From Figure 2, it doesn't look like any neurons were consistently active across the entire delay. Did any neurons significantly sustain information over the entire period? This seems relevant to the ongoing debate about sustained vs. transient working memory representations in non-human primates.

5. Related to the above point, the sequential activation of neurons seems consistent with a synfire chain model of working memory. This is generally what is found in other, non-primate, mammals. For example, you see sequential activation during memory time periods in mice (e.g., work from Carl Petersen or David Tank's groups). This is in contrast to the classic 'attractor' based models of working memory in primates. I think this is an important point to discuss, as it could change the interpretation of the results (although I would expect interference to still disrupt sequence generation in synfire chains).

6. Neurons in Figure 3 seem to respond during the delay period in time windows of ~250 ms. Is this intrinsic to the neural response or is it related to the 200 ms smoothing window that was used? In other words, would a similar pattern be observed if smaller smoothing windows were used?

7. In the paragraph starting on line 396 the authors argue that the existence of information on error trials, even on high load trials, suggests that memory is not an 'all-or-none' phenomena. With this, they argue against slot models of working memory. However, it seems like this was only true for the sample period – there is no information during the memory delay (Figure 3B). So, couldn't one interpret these results as an inability for a sample stimulus to be sustained during memory – in other words, that it didn't make it into a 'slot'? To be clear, I'm not arguing for the slot model, I would just suggest tempering the interpretation of this result.

Specific Comments

1. Figure 3A shows clustering of neurons by response profile. It seems odd that the second and third to last groups are not ordered by when their response occurred in time. In other words, I would have naively expected the second to last cluster to be closer to the fourth to last cluster (as they have more similar time courses). Am I missing something?

2. In Figure 5B the authors measure the DNR effect for neurons that gain information with increasing load. In the legend, they state N = 8 neurons for the delay period. This is much lower than the number of selective neurons shown in Figure 3C. What causes the discrepancy? Also, is N=8 significantly more neurons than expected by chance?

3. Line 171 seems like it has a typo – I assume it is a high-pass filter of 500 Hz and low-pass of 7125 Hz?

4. On line 256, in the legend for Figure 3 – my math may be wrong, but aren't there 36 neurons with selectivity in load 1?

5. On line 535, the authors reference Lebedev et al., 2004 as evidence for attention and WM overlapping in PFC of monkeys. My reading of that manuscript is that they argue for general separation of the two functions (with minimal overlap).

*Reviewer #3 (Recommendations for the authors):*

The paper is very well written. I am not an expert in the neural computational side of things, but otherwise have only a few queries. Mostly my comments are meant to improve the readability of the paper.

1. As you say, interference between memory representation is a source of information loss. A 2-sec ITI, I would think, generates a considerable amount of proactive interference. Thus is it possible that the neural outcomes are being driven by high levels of proactive interference generated by the current design?

2. Lines 138-140 I think could use a bit of elaboration, maybe in the form of an example. What threw me was the point about pairs of colors being locked to a position for a session. Eventually I came to understand what was meant, but the understanding needs to come earlier for the reader. Perhaps providing an example, using Figure 1, would help. For example, you could say... "Take the case shown in Figure 1. The colors blue and orange are locked to the left center position for that session. From trial to trial either may appear during the hold gaze period. If the left center position is chosen as the location of the target on that trial, then as shown in the Figure, the color will change to the other color of the assigned pair during the indicate change position."

3. It needs to be made more clear that loads 1, 2, and 3 refer to the number of ipsilateral items present on the display. This does become more clear as I read the manuscript but I was initially thrown by the fact that when I average across rows (aka loads) in Figure 1 the number didn't come out quite to what is mentioned on line 277. I assume the reason is that Figure 1 just shows a whole number averages.

4. Figure 2 needs clarification. First, on the x-axis, the label of "sample" is misleading (also appears throughout the text, eg line 305). I believe it represents the "hold gaze" period so it should say that. Alternatively, in Figure 1A, the "hold gaze" label in the second panel could say "sample". In fact, those labels in Figure 1A (center gaze, hold gaze, delay, indicate change) are more description of what happens in those periods rather than names of the periods, which could be: start period, sample period, delay period, response period.

5. Second (continuing on with Figure 2), it took a good deal of effort to realize what was meant by color 1 and color 2. This comment takes me back to my comment #2. regarding lines 138-140. I eventually realized that color 1 and color 2 refer to the pair of colors that appear in a hold-gaze period when that particular position is selected as the target position. Because the label "sample" was unclear, I had thought that color 1 was one of the colors during the hold-gaze period and color 2 was the other color that appeared during the indicate-change period.

6. Third (still on Figure 2). Once comments 4 and 5 were sorted then I realized that load 1, 2, and 3 (blue-yellow-red) all represent the SAME stimulus pair but with different associated loads. I was thrown because the colors (blue-yellow-red) of load 1, 2, and 3 are different, and so it made me think that color 1 and color 2 across load 1, 2, and 3 referred to different colors as well. It's all obvious now, but the issues in comments 4, 5, and 6 created a perfect storm of confusion for a while. I don't mind the three colors anymore, but as always clarification would help.

---

## [Author Response]

Essential revisions:The reviewers were each positive about the research and the paper. In reviews and discussion, a number of issues were raised which the authors should address in their revised manuscript. Below are the most important points, which are considered essential revisions. The individual reviewer comments with additional detail are appended to this list for the authors' reference.1. Load-dependent increases in color tuning seem counter to many predictions, and the authors should clarify how the DNR model can explain an increase in information as the number of stimuli increase. The suggestion appears to be that, if neurons carry more information for load 2 trials compared to load 1 trials, then this reflects an interaction between the two items that increases the information signaled, but how exactly this might occur isn't clear. There was also confusion about how non-linear effects on neural responses, which are common in the literature, relate to the DNR model. The reviewers felt that a slope near 0 is not sufficient evidence to argue conclusively that the results can be explained by DNR. It would be helpful to consider other evidence in the data and/or other normalization mechanisms, and provide a specific example neuron that shows an increase in information, along with its response to the sample, reference, and paired stimuli.

We apologize for the apparent lack of clarity in our discussion of the data. We did not mean to imply that DNR, as such, can explain the information gain at load 2. In fact, the computational principle of DNR (Heeger, 1992; Carandini and Heeger, 2013) is in tune with a loss of information at higher loads. To account for the increase in information some neurons show with increasing load we refer to a variation / application of DNR, the ‘normalization model of attention’ (Reynolds and Heeger, 2009). The model incorporates DNR to explain observations (in visual cortices) under different forms of attention. At its core are attentional fields and gain factors that affect neuronal responses. In this model we propose that DNR can explain the effect of information gain at higher loads (and the flattened linear fit Figure 5B) under the added assumption of an attentional process.

Our interpretation is based on the following: A neuron’s response (firing rate) to color 1 and color 2 (each presented by itself at the ‘favorite location’ / load 1) may be uninformative (similar firing rate for both colors). Presenting an additional color (increasing the load to 2) should affect the neurons firing rate based on DNR. As is the case for neurons that lose information, normalizing the firing rate should reduce the firing rate difference. In order to understand why we find an increase in the difference we lean on attentional modulation (normalization model of attention). If an attentional factor was present then this might have affected the DNR to become unequal (i.e., attention to the color at the additional location differentially affects firing rate for color 1 at the favorite location and color 2 at the favorite location).

Take for example the neuron in Figure 1 (Figure 5—figure supplement 2). The firing rate for the brown and blue square at the middle right position is very similar (the load 1 condition). When adding a green square at the bottom right location (the load 2 condition) the firing rate now strongly differentiates between brown and blue.

This effect can be detected on the population level and subsequently be interpreted under the normalization model of attention. If the slope of fit of the SE/SI index deviates from 0.5, this indicates an unequal interaction between colors. Applied to the example above, we’d argue that this fits an attentional effect elicited by the green square that creates the neuronal differentiation between colors at load 2, where there was none at load 1.

We chose to analyze the effect at this more abstract population level, because it offers a good comparison between the different subpopulations of neurons that lose and gain information (also see remarks to question (6) below). Furthermore, an analysis of individual interactions was not feasible due to the many different possible combinations that would have required many more trials than were available. Thus, while we think that our data fits this interpretation, we also tried to make the point that we cannot explicitly test for an attentional effect, just that our observed data (the neurons that gain information) would, nonetheless, fit an explanation based on an attentional process.

We realize now that the shift from the fundamental divisive normalization computation to the derived normalization model of attention may not have been transparent enough in our manuscript and added more detail and explanations to the result section and discussion.

The new section in the Results section (lines 300 ff.) reads:

“The ‘normalization model of attention’ (Reynolds and Heeger, 2009) incorporates divisive normalization, and can explain how attention can modulate neuronal responses. By attending a preferred (or non-preferred) second colored square in the load 2 condition the neuronal response of a neuron to the target location (i.e., to color A and to color B at the favorite location) might be altered. As a result, a difference between color A and B may arise even though each color by itself elicited a similar response. In other words, the interaction between the additional color and the target color is unequal. Neurons without a color differentiation at load 1 that gained differentiation at load 2 through this process (e.g. if the interaction of probe color A with reference color A is larger than the interaction of probe color A with reference color B, see Figure 5—figure supplement 2 for an example), should have a population regression slope smaller than 0.5.”

The new section in the discussion (lines 401 ff.) reads:

“Finally, the DNR computation may explain the responses of the neurons that gained information at load 2 through attentional processes predicted by the ‘normalization model of attention’ (Reynolds and Heeger, 2009). This may appear counter-intuitive and contradictory, considering that the same process is also responsible for the loss of information. However, when attention is overtly directed to a specific (preferred or non-preferred) item within the receptive field of a neuron, the DNR computation shifts its weighting of the normalized response towards the response of the attended item (Reynolds et al., 1999; Reynolds and Heeger, 2009). This weighted normalization can produce a difference in the neuronal response to both color identities at load 2, even if the neuronal response was non-informative at load 1. At the population level we were able to observe such an effect as the reduced slope of the selectivity/interaction fit. Thus, an attentive process might have enhanced information in WM at higher loads.

It is important to clarify that, as we did not use any form of attentional cueing in our study, we cannot explicitly test for such an attention effect. However, we do know that the animals participating in this study can use attentional cues to enhance their WM (Fongaro and Rose, 2020). The attention cues used by (Fongaro and Rose, 2020) positively affected not only encoding but also the maintenance and retrieval of the information held in WM, comparable to results from monkeys and humans (Brady and Hampton, 2018; Souza and Oberauer, 2016). We, therefore, want to emphasize that our data is in line with the interpretation that the birds possibly attended a load 2 stimulus array differently than a load 1 stimulus array in order to enhance their performance in trials with higher loads.”

2. It should be more clearly stated when working memory loads refer to the number of ipsilateral items on the display, particularly in the section on single unit analyses and in Figure 1.

We apologize for our confusing use of the terms ipsilateral and contralateral. We added clarifications. In all behavioral analyses the terms ipsi- and contralateral refer to the location of the change. Here we aimed to examine the likelihood of detecting a change depending on the load. We found no main effect of the number of items on the opposite side.

In the neural analysis the terms ipsi- and contralateral refer to a neurons’ favored location. Here we were interested to investigate the effect of additional squares on the amount of color information at the favorite location. We found a stronger modulation by additional squares in the same visual hemifield and restricted most analysis to one side (i.e. load 1-3) to reduce the number of trial combinations.

We have clarified this in the manuscript (lines 586 ff.):

“All correct trials were included in the analysis of neural data. Depending on the analysis we refer to different ‘load conditions’ relative to referential sides of the screen which are either ipsilateral (same side) or contralateral (opposite side), each with a possible load between one and three items. For the behavioral analyses the terms ipsilateral and contralateral refer to the location of the change that had to be detected. For the neurophysiological analyses the terms ipsilateral and contralateral refer to the respective neuron’s favorite location (described in the following section).”

We have also added the clarification to the figure captions of Figures1, 2 and 4:

Figure 1: (B) Boxplot of performance for different ipsilateral loads (i.e., on the side where the change occurred).

Figure 2: Color discrimination in the neuronal response (information, PEV) generally decreases with load, but some neurons show the opposite effect. Shown are the three ipsilateral load conditions (i.e., load increases on the same side as the neuron’s favorite location). Ipsilateral loads are one (blue), two (yellow), and three (red). The labels ‘color A’ and ‘color B’ always refer to the same pair of colors at the neuron’s favorite location, irrespective of the load condition.

Figure 4: Information encoding at the population level. (A) Color information (PEV) decreases with an increasing ipsilateral load (i.e., on the same side as the neuron’s favorite location) but not with an increasing contralateral load (i.e., on the opposite side to the neuron’s favorite location).

3. More clarity on the task design, terminology, and figures is needed. Specifically, what it means that color pairs are "fixed" to some of the locations, and how colors are designated "color 1" and "color 2" and whether these designations are constant across loads 1 – 3 (Figure 2) should be clarified. Perhaps an example would help here. In addition, more information on the perceptual similarity (to crows) of the colors used, and how this did or did not impact performance is important. In Figures 1 and 2, the labels "hold gaze" and "sample" appear to be used interchangeably, leading to some confusion, and consistent terms would be helpful.

We want to apologize for the unclear task description and for the inconsistent labelling of task phases that has created so much confusion.

We added a concrete example to explain the fixed locations and color changes between the two possible colors at a location (lines 531 ff.):

“The stimuli were presented at six fixed locations on the screen (1 – 6, Figure 1A). In each session, one pair of colors was assigned to each of the six locations. Each location had its own distinct pair. These pairs were randomly chosen from a pool of 14 colors (two color-combinations were excluded since the animals did not discriminate them equally well during a pre-training). Let us consider figure 1A as an example. The color-change occurs in the middle-left where turquois (T) is presented during the sample and orange (O) during the choice. In this particular session, the middle-left could thus show either of the following colors during the sample and choice: T-O (shown in Figure 1A); O-T; O-O; T-T; None-None. On the next session a new random pair of colors was displayed at this location.”

“For identification and analysis an arbitrary label was assigned to each of the randomly drawn colors at the start of each session (i.e., left middle location: ‘color A’, or ‘color B’; left top location: ‘color A’, or ‘color B’; etc.). These indices do not refer to the order of presentation of the colors at any time but were held constant for neuronal analysis. The order of presentation of colors within a pair, the target location (where the color change occurred), and the number of stimuli in the array (two to five) were randomized and balanced across trials so that each condition had an equal likelihood to appear.”

We have updated figure 1A and replaced the ‘hold gaze’ and ‘indicate change’ labels with the name of the phase that we use throughout the rest of the manuscript (i.e., sample and choice, respectively). We further added an introductory sentence to the figure caption briefly describing the task:

Figure 1: (A) Behavioral paradigm (reproduced from Balakhonov and Rose, 2017). The birds had to center and hold their gaze for the duration of the sample and delay period, and subsequently indicate which colored square had changed.

Color vision in in birds is based on 4 cone types, i.e., three cone types similar to the mammalian cones and one additional that has peak absorption in the UV spectrum around 360 nm (Kelber, 2019). Thus, birds have excellent color vision and exceed primate color vision. To ensure that specific color pairs did not influence change detection, the color pairs had been chosen for change detection based on initial training of the task, so that the animals had good change detection rates across all possible color pair combinations (two possible color pairs were excluded from use due to color similarity, see also Balakhonov and Rose, 2017).

4. A major strength of the paper is cross-species comparisons, and the reviewers suggest that the discussion of these comparisons be expanded beyond monkey neurophysiology. For instance, how do the current findings relate to sequential activity commonly reported in non-primate mammals (versus the more stable 'attractor' dynamics in primate models of working memory)? How might this work relate to behavior and neuroimaging in humans that is suggestive of DNR?

We added a short section in the discussion focused on comparisons to synfire chain models in rodents that refer to Rajan et al., 2016 and Harvey et al., 2012 (lines 446 ff.):

“This may be reconcilable with some other contemporary models of WM. One major type of those models implements ‘synfire chains’, where individual neurons fire sequentially (and transiently) to bridge temporal gaps and maintain task relevant contents (Rajan et al., 2016). This has, for example, been reported to be the case in posterior parietal cortex of mice performing a T-maze task that required WM for cued spatial locations to be maintained (Harvey et al., 2012). The transient activity of neurons that we report (Figures 2 and 3A) might fit into such models. However, our results can only be compared very cautiously to this (since even small changes in task design significantly alter neuronal responses, e.g. Lara and Wallis, 2014).”

We have added a short section concerning human neuroimaging studies to the discussion (lines 342 ff.):

“This suggests that WM could be conceptualized as a continuous resource that has to be divided between the two items (Bays and Husain, 2008; Berg et al., 2012; Wilken and Ma, 2004), rather than two ‘simple’ slots that would each have the same amount of information irrespective of the memory load. This is also consistent with results of human neuroimaging that report decreased signal amplitude and precision with increasing memory load (Emrich et al., 2013; Sprague et al., 2014).”

5. The statistical corrections for multiple comparisons across time and/or groups should be clarified, particularly in the single unit analyses.

We have made clarifications in the manuscript concerning the following issues.

We performed all analyses on which we base our interpretations (i.e., inter-load effects, correct vs. error trials, DNR) on p-values corrected for multiple comparisons (this is stated throughout the Results section).

Lines 586-592 and 609-610:

“We did not test for all six possible locations, we restricted analysis to the three locations on the right side of the screen. Of these three locations, we chose the location that had the cumulative highest PEV across the sample period (i.e., the favorite location), which for this purpose was segmented into four 200 ms bins. Significant PEV values for individual neurons were then calculated for the favorite location in 200 ms bins using permutation testing at an α of 0.05.”

Lines 618-626:

“We did not use the number of significant neurons as a measure for information about color, but rather their effect size, which is a much more derived statistical value for which there is no universally accepted critical threshold (e.g., see the small but still very relevant values of Buschman et al., 2011). As we were looking for the effect of load on WM within NCL we aimed to strengthen our analysis by including as many neurons as possible. Therefore, to determine if a single neuron had a significant amount of information about color, we did not apply further statistical corrections beyond the permutation testing.”

We hypothesized that information would decrease as WM load increases, which should be an effect happening at the level of the neuronal population, and not necessarily (as we saw) at the level of individual neurons. If we were to only include those neurons that had the most information at the individual loads (i.e., those with the highest PEV values that are significant even under very stringent statistical criteria) we would have artificially inflated the amount of information present at each load. By including neurons that encoded less information too (at an uncorrected p-value of 0.05) our analysis population was more resistant to such outlier effects.

Nonetheless, we can also report that when we apply a more stringent criterion for significance (e.g., two consecutive non-overlapping bins, i.e., a temporal requirement for consistency of the effect). When we do so, all reported effects retain their quality (see Figure 2, (Figure 4—figure supplement 2) compare to Figure 4B in the manuscript). The flipside of this is that the number of ‘significant’ neurons decreases, while the amount of information about color (PEV values) increases, which is to be expected, because the selection process of neurons now retains only those neurons showing the strongest effects (highest PEV). We think (for the reasons mentioned above) that this would represent a case of over selection (‘cherry picking’) from the obtained data, which is why we would like to retain our original population.”

6. How do the authors reconcile the fact that the number of single neurons tuned to color tends to increase at higher loads, but there is a loss of color information at the population level?

There seems to be a misunderstanding. The number of single neurons with significant color information in the three load conditions is quite stable. Author response table 1, shows this (values are taken from the pie charts of Figure 3C)

**Author response table 1. sa2table1:** Proportion of neurons with significant color information (number of neurons) at the different load conditions in the sample and the delay phase.

Phase	Load 1	Load 2	Load 3
Sample	55 % (137 neurons)	55 % (137 neurons)	43 % (107 neurons)
Delay	36 % (34 neurons)	37 % (35 neurons)	39 % (37 neurons)

The overall loss of information at the population level (the central finding of this study) and each neuron’s individual significance are not necessarily related. This is because the absolute amount of information about color (i.e., the PEV value) at load 2 or 3 can be lower than at load 1 while still remaining significant. We purposefully investigated and compared these significant subpopulations to investigate the central question of information loss and capacity. We expand on how neurons lose information and why some neurons have significant information at higher loads when they did not have this color information at load 1.

7. Reviewers requested more detail on whether any neurons in the population sustained information across the delay, since it is relevant to the ongoing debate about sustained vs. transient working memory representations in non-human primates.

There were only few neurons that sustained an elevated firing rate throughout longer phases of the delay period. Those neurons did not differentiate their firing rate in relation to the different colors, i.e. they did not carry significant amounts of color information throughout their sustained activity. This can be observed in the cluster plot (Figure 3A) where it is obvious that information was ever only transiently maintained by individual neurons. This is also reflected in the example neurons shown in Figures 2, S4 and S5. Insofar our results are consistent with contemporary conceptions of WM (in primates), that suggest that persistent delay activity may not be the central mechanism by which information is maintained in WM (e.g, Miller et al., 2018, Lundqvist et al., 2018). We have added a short section in the discussion to address this point, together with our remarks about synfire chains (see point 4). (lines 442 ff.)

“There is also ongoing debate about the role of sustained activity during delay periods and how it relates to WM (Constantinidis et al., 2018, Lundqvist et al., 2018, Miller et al., 2018). We cannot report of any neuron that showed persistent activity comparable to those reported by classical WM studies in PFC (e.g., Fuster and Alexander, 1971, Funahashi et al., 1992), or in NCL (Diekamp et al., 2002, Veit and Nieder, 2013, Veit et al., 2014).”

8. Neurons in Figure 3 seem to respond during the delay period in time windows of ~250 ms. Is this intrinsic to the neural response or is it related to the 200 ms smoothing window that was used? In other words, would a similar pattern be observed if smaller smoothing windows were used?

The neuronal response is not dependent on the window size. Using a window size of 100 ms or 50 ms returns virtually the same response. The raster plots at the top of each neurons’ figure give an indication of that. We have included in Author response image 1, the example neuron of figure 2A smoothed with a 100 ms window.

**Author response image 1. sa2fig1:** Example neuron (same as in Figure 2A of the manuscript), smoothed with 100 ms bins. Top: raster plot, where every dot represents a single spike during the individual trials (rows of dots); middle: peri-stimulus-time histogram (PSTH) of average firing rate (solid line for color ID 1, dashed line for color ID 2) with the standard error of the mean (shaded areas); bottom: percent explained variance of color identity (a measure of information about color) along the trial, the line at the top of the y-axis indicates significant bins.

9. The authors argue that the existence of information on error trials, even on high load trials, suggests that memory is not an 'all-or-none' phenomena, and this is inconsistent with the slot models of working memory. However, it seems like this was only true for the sample period, and there was no information during the memory delay (Figure 3B). Could this be interpreted as an inability for a sample stimulus to be sustained during memory – in other words, that it didn't make it into a 'slot'?

We have included this point in the Results section (lines 232 ff.):

“A possible alternative ‘slot-model’ explanation would be that, on error trials, the color information was completely lost after the sample phase, because it was not successfully transferred into a slot (or that a slot was not available to take on information). The graded amount of information on correct trials is not compatible with the simple (all or none) slot model but could fit the ‘slots and averaging model’ (Zhang and Luck, 2008).”

10. While the results are conceptually consistent with classic effects in monkey lateral PFC, Lara and Wallis (Nat Neurosci, 2014) reported a near absence of color tuning in a very similar color change detection task, and instead found predominantly spatial attention signals. Can the authors discuss this discrepancy?

We have added a short paragraph discussing the discrepancies to Lara and Wallis (2014) (lines 359 ff.):

“We probed the WM capacity of crows using colored squares, based on the task design of Buschman et al., (2011). Using the identical task allowed us to directly compare our neuronal results of WM capacity from NCL to results from PFC of monkeys. Task similarity is very important for such cross species comparisons as even small changes in task parameters may introduce substantial differences in neuronal responses, leading to potentially different conclusions. In a task similar to the one used here Lara and Wallis (2014) have found that neurons in the PFC of monkeys encoded nearly no information about color, but instead about location. In their task monkeys had to memorize the color of squares at two locations on a screen, and were again confronted with a colored square at one of the two locations after a delay. The monkeys then had to indicate if the color at that location had changed. Lara and Wallis (2014) discuss the absence of color information in the neurons they recorded in relation to the task of Buschman et al., (2011), who like us, did find color information. In brief, the exact task design may determine the neuronal encoding of task relevant information (Lara and Wallis, 2014). Similar to the complex contribution of PFC neurons to WM, neurons of NCL can also encode a wide range of very different task relevant aspects, like color (this study), spatial locations (Veit et al., 2015, Rinnert et al., 2019), and more abstract items like rules (Veit and Nieder, 2013), and numerosities (Ditz and Nieder, 2015).”

11. Interference between memory representation is mentioned as a source of information loss, and a 2-sec ITI likely generates a considerable amount of proactive interference. Thus is it possible that the neural outcomes are being driven by high levels of proactive interference generated by the current design?

We agree that proactive interference could be a source of overall difficulty in our task and effect, for instance, the overall estimate of WM capacity.

However, we think that it is highly unlikely that proactive interference affected the reported results in any meaningful way since trial-types were randomized within each session.

It may be also noteworthy that the 2s ITI does not reflect the entire separation between trials. Birds were given additional 2 seconds to consume their reward after correct choices and received a 10 s timeout after incorrect choices. Furthermore, the birds were free to assume head fixation on their own time, which resulted in a random offset in each trial.

Reviewer #1 (Recommendations for the authors):1. While the results are conceptually consistent with classic effects in monkey lateral PFC, Lara and Wallis (Nat Neurosci, 2014) reported a near absence of color tuning in a very similar color change detection task, and instead found predominantly spatial attention signals. Can the authors discuss this discrepancy?

Thank you for pointing out the study. Please refer to point (10) of the essential revisions.

2. Is there much known about color vision in crows? Specifically, are there any subsets of colors used in this study that could have appeared more perceptually similar/different to the crows, and does that affect performance?

Color vision in in birds is based on 4 cone types, i.e., three cone types similar to the mammalian cones and one additional that has peak absorption in the UV spectrum around 360 nm (Kelber, 2019). Thus, birds have excellent color vision and exceed primate color vision. To ensure that specific color pairs did not influence change detection, the color pairs had been chosen for change detection based on initial training of the task, so that the animals had good change detection rates across all possible color pair combinations (two possible color pairs were excluded from use due to color similarity, see also Balakhonov and Rose, 2017).

3. It wasn't immediately obvious whether the single unit analyses that looked at effects of working memory load were based on ipsilateral, or total load. Based on the behavioral results, it seems that it should be ipsilateral only, but can the authors clarify this?

We apologize for our confusing use of the terms ipsilateral and contralateral. We have addressed the issue under point (2) of the essential revisions, and added clarifications to the manuscript (Lines 586 ff. and 609 f.) and to the figure captions of Figures 1,2 and 4.

4. How do the authors reconcile the fact that the number of single neurons tuned to color tends to increase at higher loads, but there is a loss of color information at the population level?

We are not sure what the reviewer is referring to specifically. There may be a misunderstanding. The number of single neurons with significant color information in the three load conditions seems quite stable. (Author response image 1) shows this (values are taken from the pie charts of Figure 3C)

The overall loss of information at the population level (the central finding of this study) and each neuron’s individual significance are not necessarily related. This is because the absolute amount of information about color (i.e., the PEV value) at load 2 or 3 can be lower than at load 1 while still remaining significant. We purposefully investigated and compared these significant subpopulations to investigate the central question of information loss and capacity. We expand on how neurons lose information and why some neurons have significant information at higher loads when they did not have this color information at load 1 (also see below).

5. Load-dependent increases in color tuning also seem counter to many predictions. Are there any similar results in the NHP literature?

We are not aware of any studies reporting such gains of information.

Also, I wasn't able to follow how divisive normalization results in gain of information with increasing load. Can the authors elaborate on this?

We apologize for the lack of clarity in discussing this aspect of our data. Please refer to point (1) of the essential revisions, where we have explained the effect and included an example.

6. Is it predicted that neurons without clear color tuning also exhibited load effects consistent with divisive normalization (Supp Figure 7)? If not, how is this result interpreted?

Yes, DNR should also occur if significant differentiation between colors (i.e., color tuning) is not present. This is because the computation is emerging from the activity of neuronal network (populations) irrespective of tuning of individual neurons. The point is that those neurons (SFigure 7) weren’t affected by the conjectured attentional processes. This is relevant insofar that we chose to split the population of neurons into three groups based on their significances for color information (i.e., (1) information at load 1 and subsequent loss of information at higher load; (2) no information at load 1 but gain of information at load 2; (3) no information at load 1 and no gain of information at load 2) and expected to find group specific results. Namely, classic DNR reducing information (due to equal interaction, i.e. slopes ~ 0.5) for group (1) and (3) and unequal interaction due to attention resulting in slopes different from 0.5 for group (2). Which is what we found.

Reviewer #2 (Recommendations for the authors):General Comments1. I think my biggest concern is regarding the argument for divisive normalization like regularization (DNR), particularly for neurons that have increased information for a memory load of 2 items. The authors seem to suggest that if neurons carry more information for load 2 trials compared to load 1 trials, then this reflects an interaction between the two items such that information increases. First, how exactly the authors are envisioning this isn't clear to me. It would be helpful to provide a specific example neuron that shows an increase in information, along with its response to the sample, reference, and pair stimuli.That being said, if I understand correctly, the authors seem to imply there is a non-linear effect on the neural response of some neurons when two stimuli are presented. This seems consistent with previous work (e.g., non-linear responses seen by Fusi). However, this doesn't seem consistent with the DNR model. The DNR model makes a fairly clear prediction that the response to two stimuli will be a (weighted) average of the response to each stimulus alone. Given this, it seems to me that the DNR model doesn't have a mechanism for explaining an increase in information as the number of stimuli increase. Typically, in the DNR framework, a slope close to zero is taken as evidence that the response to the Pair is equal to the Ref (e.g., when attention is shifted to the reference). That seems unlikely to be the case here, but highlights how a slope near 0 is not sufficient to argue that the results can be explained by a DNR effect.Perhaps I am missing something, in which case I would suggest the authors make this point more clearly in the current manuscript. Or, if the classic DNR model can't predict the increase in information for two-item preferring neurons, then I think the current results could argue for other normalization mechanisms (or a combination of mechanisms) that could be acting in the brain.

We apologize for the lack of clarity in discussing this aspect of our data. Please refer to point (1) of the essential revisions, where we have explained the effect and included an example.

2. It wasn't clear to me what statistics were corrected for multiple comparisons across time and/or groups. To measure sensory/memory information in individual neurons, the authors used a percent explained variance measure to test if the neuron responded differently to two different colors. As far as I can tell, this test was performed for each of six different stimulus locations and at multiple points in time (at least four). It isn't clear from the text whether the authors corrected for these multiple comparisons when determining whether a neuron was significant.

We apologize for the lack of clarity in our methods. We have added clarifications to the manuscript (lines 586-592 and 609-610). Please refer to our answer to point (5) of the essential revisions for a detailed account of our statistical methodology.

3. As noted above, the current work is consistent with behavior and neuroimaging in humans. In particular, there has been evidence from human neuroimaging work arguing for divisive normalization in working memory (e.g., Sprague et al., 2014) that seems relevant and worth citing.

Thank you for pointing this out, we have added this suggestion to the discussion (lines 342 ff.):

“This suggests that WM could be conceptualized as a continuous resource that has to be divided between the two items (Bays and Husain, 2008; Berg et al., 2012; Wilken and Ma, 2004), rather than two ‘simple’ slots that would each have the same amount of information irrespective of the memory load. This is also consistent with results of human neuroimaging that report decreased signal amplitude and precision with increasing memory load (Emrich et al., 2013; Sprague et al., 2014).”

4. On line 310 the authors state that "most neurons did not sustain information about color (…) throughout the entire sample or memory delay". From Figure 2, it doesn't look like any neurons were consistently active across the entire delay. Did any neurons significantly sustain information over the entire period? This seems relevant to the ongoing debate about sustained vs. transient working memory representations in non-human primates.

Thank you for mentioning this topic. Discussing these aspects of neuronal response, also in relation to other relevant models of mammalian WM may help to round out our comparative perspective (see also answer to the next point). We have added a short section in the discussion to address this point (lines 442 ff.). Please refer to our answer to point (7) of the essential revisions.

5. Related to the above point, the sequential activation of neurons seems consistent with a synfire chain model of working memory. This is generally what is found in other, non-primate, mammals. For example, you see sequential activation during memory time periods in mice (e.g., work from Carl Petersen or David Tank's groups). This is in contrast to the classic 'attractor' based models of working memory in primates. I think this is an important point to discuss, as it could change the interpretation of the results (although I would expect interference to still disrupt sequence generation in synfire chains).

We added a short section in the discussion focused on comparisons to synfire chain models in rodents that refer to Rajan et al., 2016 and Harvey et al., 2012 (lines 446 ff.). Please refer to our answer to point (4) of the essential revisions.

6. Neurons in Figure 3 seem to respond during the delay period in time windows of ~250 ms. Is this intrinsic to the neural response or is it related to the 200 ms smoothing window that was used? In other words, would a similar pattern be observed if smaller smoothing windows were used?

The neuronal response is not dependent on the window size. Using a window size of 100 ms or 50 ms returns virtually the same response. The raster plots at the top of each neurons’ figure give an indication of that. We have included an example under point (8) of the essential revisions.

7. In the paragraph starting on line 396 the authors argue that the existence of information on error trials, even on high load trials, suggests that memory is not an 'all-or-none' phenomena. With this, they argue against slot models of working memory. However, it seems like this was only true for the sample period – there is no information during the memory delay (Figure 3B). So, couldn't one interpret these results as an inability for a sample stimulus to be sustained during memory – in other words, that it didn't make it into a 'slot'? To be clear, I'm not arguing for the slot model, I would just suggest tempering the interpretation of this result.

Thank you for raising this point. Please refer to our answer to point (9) of the essential revisions.

Specific Comments1. Figure 3A shows clustering of neurons by response profile. It seems odd that the second and third to last groups are not ordered by when their response occurred in time. In other words, I would have naively expected the second to last cluster to be closer to the fourth to last cluster (as they have more similar time courses). Am I missing something?

The specific clustering of the latter groups is indeed peculiar, it is, however based on the algorithm’s classification that takes the entire trials’ activity into account. The third to last group is a sibling to the fourth to last that seems to have two centers of peak information (one at the beginning of the delay and one smaller at the end). The second to last group does not have the peak in information at the end of the delay. This likely affected grouping into the observed pattern. This may be resolved by changing the target number of clusters, but would then be suboptimal in terms of the optimal number of clusters as determined by the within-cluster sum-of-squares measure.

2. In Figure 5B the authors measure the DNR effect for neurons that gain information with increasing load. In the legend, they state N = 8 neurons for the delay period. This is much lower than the number of selective neurons shown in Figure 3C. What causes the discrepancy? Also, is N=8 significantly more neurons than expected by chance?

That particular subgroup only has 8 neurons (nonsignificant at load 1 and significant in the delay (i.e., the entire delay measured as one bin) at load 2). The values of significant neurons between Figure 3C and 5 are not the same because 3C states neurons with significant information binned in 200 ms bins (see line 608 ff. of the methods), whereas 5C states neurons with significant information in the sample and/or delay as one bin (800 ms and 1000 ms) respectively (see lines 676 ff. of the methods).

The reason for pooling neurons across a larger time window for the analyses concerning Figure 5 was to facilitate interpretation of the results. By considering the entire sample or delay phase we were able to interpret the effect at the level of the population response irrespective of individual neurons’ specific response curves.

We have added clarifications to the figure caption of Figure 5:

(A) Information carrying neurons in the sample phase (as one bin; n = 105; left) and delay phase (as one bin; n = 43; right) population. (B) Information gaining neurons in the sample phase (as one bin; n = 56 ; left) and delay phase (as one bin; n = 8; right) population. The red line indicates the regression fit.

And have added our reason for changing the criterion for significance in the methods section (lines 679 ff.):

“We considered the entire sample and delay phase because we wanted to analyze the population response as a whole, irrespective of highly diverse response profiles of individual neurons.”

3. Line 171 seems like it has a typo – I assume it is a high-pass filter of 500 Hz and low-pass of 7125 Hz?

Yes, this has been corrected.

4. On line 256, in the legend for Figure 3 – my math may be wrong, but aren't there 36 neurons with selectivity in load 1?

The number of neurons seems to be correct. Significant delay neurons, at load 1 are: 28 % (only load 1) + 3 % (load 1 and load 2) + 5 % (load 1 and load 3) = 36 % of 94 delay neurons in total: 0.36 * 94 = 33.84, adjusted for rounding errors (as depicted numbers are rounded) = 34 neurons, as stated.

5. On line 535, the authors reference Lebedev et al., 2004 as evidence for attention and WM overlapping in PFC of monkeys. My reading of that manuscript is that they argue for general separation of the two functions (with minimal overlap).

Lebedev and colleagues find three types of neurons: memory neurons, attention neurons and hybrid neurons (encoding both memory and attention). They also explicitly state that there is a likely overlap between the different functions and that a purely mnemonic function of PFC is too simple to explain the data. We therefore think it is a good example of the literature (backed up by more novel studies, like Panichello and Buschman, 2021) to illustrate the co-occurrence of WM and attention in PFC.

We have extended the discussion to more clearly incorporate the relationship of PFC neurons to both memory and attention (lines 387 ff.):

“Beyond the domain of sensory signals, attention and WM may be directly linked. Neuronal correlates of WM and attention overlap in PFC neurons, for example Lebedev et al., (2004) found that a substantial amount of PFC neurons encode either an attentional signal, or a memory signal, and some (hybrid) neurons do both. A purely mnemonic function of PFC thereby seems unlikely. Indeed, very recently Panichello and Buschman (2021) have reported that at the population level neurons of PFC encode ‘both the selection of items from working memory and attention to sensory inputs’ (p.2), rather than just memory content.”

Reviewer #3 (Recommendations for the authors):The paper is very well written. I am not an expert in the neural computational side of things, but otherwise have only a few queries. Mostly my comments are meant to improve the readability of the paper.1. As you say, interference between memory representation is a source of information loss. A 2-sec ITI, I would think, generates a considerable amount of proactive interference. Thus is it possible that the neural outcomes are being driven by high levels of proactive interference generated by the current design?

Thank you for raising this point. Please refer to our answer to point (11) of the essential revisions.

2. Lines 138-140 I think could use a bit of elaboration, maybe in the form of an example. What threw me was the point about pairs of colors being locked to a position for a session. Eventually I came to understand what was meant, but the understanding needs to come earlier for the reader. Perhaps providing an example, using Figure 1, would help. For example, you could say... "Take the case shown in Figure 1. The colors blue and orange are locked to the left center position for that session. From trial to trial either may appear during the hold gaze period. If the left center position is chosen as the location of the target on that trial, then as shown in the Figure, the color will change to the other color of the assigned pair during the indicate change position."

Thank you for pointing this out. We added a concrete example to explain the fixed locations and color changes between the two possible colors at a location in (lines 531 ff.). Please refer to our answer to point (3) of the essential revisions.

3. It needs to be made more clear that loads 1, 2, and 3 refer to the number of ipsilateral items present on the display. This does become more clear as I read the manuscript but I was initially thrown by the fact that when I average across rows (aka loads) in Figure 1 the number didn't come out quite to what is mentioned on line 277. I assume the reason is that Figure 1 just shows a whole number averages.

We apologize that our use of the different load conditions was not clear enough. We have added a section to clarify the terminology around the loads, and the terms ‘ipsilateral’ and ‘contralateral’ (lines 586 ff.). Please refer to our answer to point (2) of the essential revisions.

We further added a remark to the caption of figure 1 that indicates that the displayed numerical values were rounded to the nearest integer.

4. Figure 2 needs clarification. First, on the x-axis, the label of "sample" is misleading (also appears throughout the text, eg line 305). I believe it represents the "hold gaze" period so it should say that. Alternatively, in Figure 1A, the "hold gaze" label in the second panel could say "sample". In fact, those labels in Figure 1A (center gaze, hold gaze, delay, indicate change) are more description of what happens in those periods rather than names of the periods, which could be: start period, sample period, delay period, response period.

We have updated figure 1A and replaced the ‘hold gaze’ and ‘indicate change’ labels with the phase appropriate name of the phase that we use throughout the rest of the manuscript (i.e., sample and choice, respectively). We further added an introductory sentence to the figure caption briefly describing the task:

“Figure 3: (A) Behavioral paradigm (reproduced from Balakhonov & Rose, 2017). The birds had to center and hold their gaze for the duration of the sample and delay period, and subsequently indicate which colored square had changed.”

5. Second (continuing on with Figure 2), it took a good deal of effort to realize what was meant by color 1 and color 2. This comment takes me back to my comment #2. regarding lines 138-140. I eventually realized that color 1 and color 2 refer to the pair of colors that appear in a hold-gaze period when that particular position is selected as the target position. Because the label "sample" was unclear, I had thought that color 1 was one of the colors during the hold-gaze period and color 2 was the other color that appeared during the indicate-change period.

We have decided to exchange the labels ‘1’ and ‘2’ for ‘A’ and ‘B’, respectively to avoid the implication of sequential presentation and added a sentence to the figure caption to clarify the meaning of colors A and B.

Figure 4: Color discrimination in the neuronal response (information, PEV) generally decreases with load, but some neurons show the opposite effect. Shown are the three ipsilateral load conditions (i.e., load increases on the same side as the neuron’s favorite location). Ipsilateral loads are one (blue), two (yellow), and three (red). The labels ‘color A’ and ‘color B’ always refer to the same pair of colors at the neuron’s favorite location, irrespective of the load condition.

Further, refer to changes made in accordance with questions (2) and (4).

6. Third (still on Figure 2). Once comments 4 and 5 were sorted then I realized that load 1, 2, and 3 (blue-yellow-red) all represent the SAME stimulus pair but with different associated loads. I was thrown because the colors (blue-yellow-red) of load 1, 2, and 3 are different, and so it made me think that color 1 and color 2 across load 1, 2, and 3 referred to different colors as well. It's all obvious now, but the issues in comments 4, 5, and 6 created a perfect storm of confusion for a while. I don't mind the three colors anymore, but as always clarification would help.

We want to apologize for the lack of clarity and the confusion that ensued because of it. We have added the clarification concerning the colors in the figure caption (refer to answer to question (4) above). We hope all the individual changes made at the different part of the manuscript will help the readers to more easily understand the design of our task and analysis.